# The BTB-ZF gene *Bm-mamo* regulates pigmentation in silkworm caterpillars

Songyuan Wu, Xiaoling Tong, Chenxing Peng, Jiangwen Luo, Chenghao Zhang, Kunpeng Lu, Chunlin Li, Xin Ding, Xiaohui Duan, Yaru Lu, Hai Hu, Duan Tan, Fangyin Dai*

State Key Laboratory of Resource Insects, Key Laboratory of Sericultural Biology and Genetic Breeding, Ministry of Agriculture and Rural Affairs, College of Sericulture, Textile and Biomass Sciences, Southwest University, Chongqing, China

## Abstract

The color pattern of insects is one of the most diverse adaptive evolutionary phenotypes. However, the molecular regulation of this color pattern is not fully understood. In this study, we found that the transcription factor Bm-mamo is responsible for *black dilute* (*bd*) allele mutations in the silkworm. Bm-mamo belongs to the BTB zinc finger family and is orthologous to mamo in *Drosophila melanogaster*. This gene has a conserved function in gamete production in *Drosophila* and silkworms and has evolved a pleiotropic function in the regulation of color patterns in caterpillars. Using RNAi and clustered regularly interspaced short palindromic repeats (CRISPR) technology, we showed that Bm-mamo is a repressor of dark melanin patterns in the larval epidermis. Using in vitro binding assays and gene expression profiling in wild-type and mutant larvae, we also showed that Bm-mamo likely regulates the expression of related pigment synthesis and cuticular protein genes in a coordinated manner to mediate its role in color pattern formation. This mechanism is consistent with the dual role of this transcription factor in regulating both the structure and shape of the cuticle and the pigments that are embedded within it. This study provides new insight into the regulation of color patterns as well as into the construction of more complex epidermal features in some insects.

*For correspondence:
fydai@swu.edu.cn

Competing interest: The authors declare that no competing interests exist.

## eLife assessment

This **important** study identifies the gene mamo as a new regulator of pigmentation in the silkworm *Bombyx mori*, a function that was previously unsuspected based on extensive work on *Drosophila* where the *mamo* gene is involved in gamete production. The evidence supporting the role of *Bm-nano* in pigmentation is **convincing**, including high-resolution linkage mapping of two mutant strains, expression profiling, and reproduction of the mutant phenotypes with state-of-the-art RNAi and CRISPR knock-out assays. The work will be of interest to evolutionary biologists and geneticists studying color patterns and evolution of gene networks.

## Introduction

Insects often display stunning colors, and the patterns of these colors have been shown to be involved in behavior (*Sword et al., 2000*), immunity (*Barnes and Siva-Jothy, 2000*), thermoregulation (*Hadley et al., 1992*), and UV protection (*Hu et al., 2013*); in particular, visual antagonism of predators via aposematism (*Stevens and Ruxton, 2012*), mimicry (*Dasmahapatra, 2012*), cryptic color patterns (*Gaitonde et al., 2018*), or some combination of the above (*Tullberg et al., 2005*). In addition, the color patterns are divergent and convergent among populations (*Wittkopp et al., 2003b*). Due to

these striking visual features and highly active adaptive evolutionary phenotypes, the genetic basis and evolutionary mechanism of color patterns have long been a topic of interest.

Insect coloration can be pigmentary, structural, or bioluminescent. Pigments are synthesized by insects themselves and form solid particles that are deposited within the cuticle of the body surface and the scales of the wings (*Futahashi et al., 2010*; *Matsuoka and Monteiro, 2018*). Interestingly, recent studies have shown that bile pigments and carotenoid pigments synthesized through biological synthesis are incorporated into the body fluids and fill in the wing membranes of two butterflies (*Siproeta stelenes* and *Philaethria diatonica*) via hemolymph circulation, providing color in the form of liquid pigments (*Finet et al., 2023*). These pigments form colors by selective absorption and/or scattering of light depending on their physical properties (*Gürses et al., 2016*). However, structural color refers to colors, such as metallic colors and iridescence, generated by optical interference and grating diffraction of the microstructure/nanostructure of the body surface or appendages (such as scales) (*Lloyd and Nadeau, 2021*; *Seago et al., 2009*). Pigment color and structural color are widely distributed in insects and can only be observed by the naked eye in illuminated environments. However, some insects, such as fireflies, exhibit colors (green to orange) in the dark due to bioluminescence (*Oba et al., 2020*). Bioluminescence occurs when luciferase catalyzes the oxidation of small molecules of luciferin (*Syed and Anderson, 2021*). In conclusion, the color patterns of insects have evolved to become highly sophisticated and are closely related to their living environment. For example, cryptic color can deceive animals via high similarity to the surrounding environment. However, the molecular mechanism by which insects form precise color patterns to match their living environment has not been determined.

Recent research has identified the metabolic pathways associated with related pigments, such as melanins, pterins, and ommochromes, in Lepidoptera (*Tong et al., 2021*). A deficiency of enzymes in the pigment metabolism pathway can lead to changes in color patterns (*Dai et al., 2010*; *Wittkopp et al., 2002*). In addition to pigment synthesis, the microstructure/nanostructure of the body surface and wing scales are important factors that influence body color patterns. The body surface and wing scales of Lepidoptera are composed mainly of cuticular proteins (CPs) (*Liu et al., 2021*; *Yan et al., 2022*). There are multiple genes encoding CPs in Lepidopteran genomes. For example, in *Bombyx mori*, more than 220 genes encode CPs (*Futahashi et al., 2008a*). However, the functions of CPs and the molecular mechanisms underlying their fine-scale localization in the cuticle are still unclear.

In addition, several pleiotropic factors, such as *wnt1* (*Yamaguchi et al., 2013*), *Apontic-like* (*Yoda et al., 2014*), *clawless* (*Jin et al., 2019*), *abdominal-A* (*Rogers et al., 2014*), *abdominal-B* (*Jeong et al., 2006*), *engrailed* (*Dufour et al., 2020*), *antennapedia* (*Prakash et al., 2022*), *optix* (*Reed et al., 2011*), *bric à brac* (*bab*) (*Kopp et al., 2000*), and *Distal-less* (*Dll*) (*Arnoult et al., 2013*), play important roles in the regulation of color patterns.

The molecular mechanism by which these factors participate in the regulation of body color patterns needs further study. In addition, there are many undiscovered factors in the gene regulatory network involved in the regulation of insect color patterns. The identification of new color pattern regulatory genes and the study of their molecular mechanisms would be helpful for further understanding color patterns.

Silkworms (*B. mori*) have been completely domesticated for more than 5000 years and are famous for their silk fiber production (*Tong et al., 2022*). Because of this long history of domestication, artificial selection, and genetic research, its genetic background has been well studied, and approximately 200 strains of mutants with variable color patterns have been preserved (*Cheng. et al., 2003*), which provides a good resource for color pattern research. The *black dilute* (*bd*) mutant, which exhibits recessive Mendelian inheritance, has a dark gray larval body color, and the female is sterile. *Black dilute fertile* (*bd^f*) is an allele that leads to a lighter body color than *bd* and fertile females (*Figure 1*). The two mutations were mapped to a single locus at 22.9 cM of linkage group 9 (*Kawaguchi et al., 2007*). No pigmentation-related genes have been reported at this locus. Thus, this research may reveal a new color pattern-related gene, which stimulated our interest.

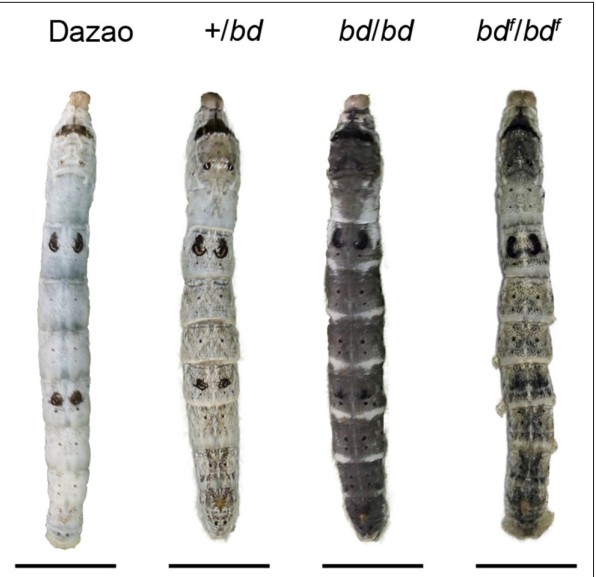

**Figure 1.** Phenotypes of *bd*, *bd*<sup>f</sup>, and wild-type Dazao larvae. The epidermis of *bd* is dark gray, and the epidermis of *bd*<sup>f</sup> is light gray at the fifth instar (Day 3) of silkworm larvae. The scale bar indicates 1 cm.

## Results

### Candidate gene of the *bd* allele

To identify the genomic region responsible for the *bd* alleles, positional cloning of the *bd* locus was performed. Due to the female infertility of the *bd* mutant and the fertility of females with the *bd*<sup>f</sup> allele, we used *bd*<sup>f</sup> and the wild-type Dazao strain as parents for mapping analysis.

The 1162 back-crossed filial 1st (BC1M) generation individuals from *bd*<sup>f</sup> and Dazao were subjected to fine mapping with molecular markers (***Figure 2***). A genomic region of approximately 390 kb was responsible for the *bd* phenotype (***Supplementary file 1–Table S1***). According to the SilkDB database (***Duan et al., 2010***), this region included five predicted genes (*BGIBMGA012516*, *BGIBMGA012517*, *BGIBMGA012518*, *BGIBMGA012519*, and *BGIBMGA014089*). In addition, we analyzed the predictive genes for this genomic region from the GenBank (***Sayers et al., 2022***) and SilkBase (***Kawamoto et al., 2022***) databases. The number of predicted genes varied among the different databases. We performed sequence alignment analysis of the predicted genes in the three databases to determine their correspondence. Real-time quantitative polymerase chain reaction (qPCR) was subsequently performed, which revealed that *BGIBMGA012517* and *BGIBMGA012518* were significantly downregulated on Day 3 of the fifth instar of the larvae in the *bd* phenotype individuals, while there was no difference in the expression levels of the other genes (***Figure 2—figure supplement 1***). These two genes were predicted to be associated with a single locus (*LOC101738295*) in GenBank. To determine the gene structures of *BGIBMGA012517* and *BGIBMGA012518*, we used forward primers for the *BGIBMGA012517* gene and reverse primers for the *BGIBMGA012518* gene to amplify cDNA from the wild-type Dazao strain. By gene cloning, the predicted genes *BGIBMGA012517* and *BGIBMGA012518* were proven to be one such gene. For convenience, we temporarily called this the *12517/8* gene.

The *12517/8* gene produces two transcripts; the open reading frame (ORF) of the long transcript is 2397 bp, and the ORF of the short transcript is 1824 bp, which is the same 5′-terminus, in the wild-type Dazao strain (***Figure 2—figure supplement 2***). The *12517/8* gene showed significantly lower expression in the *bd*<sup>f</sup> mutant (***Figure 2—figure supplement 3***), and comparative genomic analysis revealed multiple variations in the region near this gene between *bd*<sup>f</sup> and Dazao (***Supplementary file 2–Table S2***). In addition, *12517/8* was completely silenced due to the deletion of DNA fragments (approximately 168 kb) from the first upstream intron and an insertion of 3560 bp in the *bd* mutant (***Figure 2—figure supplement 4***).

To predict the function of the *12517/8* gene, we performed a BLAST search using its full-length sequence and found a transcription factor, the *maternal gene required for meiosis* (*mamo*) in *Drosophila melanogaster*, that had high sequence similarity to that of the *12517/8* gene (***Figure 2—figure***

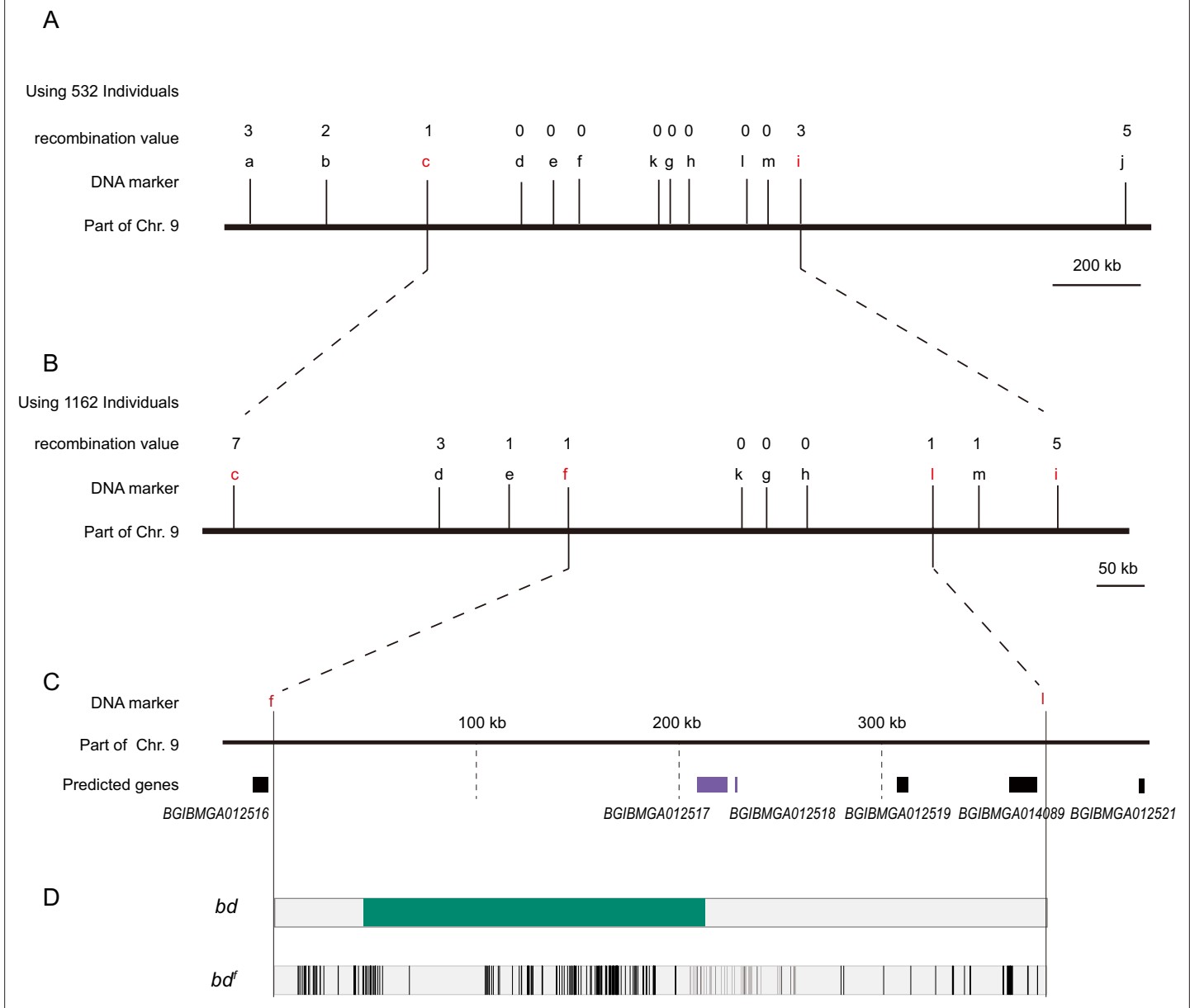

**Figure 2.** Positional cloning of the *bd* locus. (**A**) We used 532 BC1 individuals to map the *bd* locus between PCR markers c and i. The numbers above the DNA markers indicate recombination events. (**B**) A total of 1162 BC1 individuals were used to narrow the *bd* locus to an approximately 400 kb genomic region. (**C**) Partial enlarged view of the region responsible for *bd*. This region contains four predicted genes, *BGIBMGA012517*, *BGIBMGA012518*, *BGIBMGA012519*, and *BGIBMGA014089*. (**D**) Analysis of nucleotide differences in the region responsible for *bd*. The green block indicates the deletion of the genome in *bd* mutants. The black vertical lines indicate the single-nucleotide polymorphisms (SNPs) and indels of *bd*ᶠ mutants.

The online version of this article includes the following source data and figure supplement(s) for figure 2:

**Figure supplement 1.** Screening analysis of candidate genes for the *bd* mutant.

**Figure supplement 2.** Sequence analysis of the *12517/18-L* and *12517/18-S* transcripts.

**Figure supplement 2—source data 1.** The coding sequences of two transcripts of *Bm-mamo*.

**Figure supplement 3.** The expression levels of *12517/18* in the wild-type Dazao strain and *bd* allele mutants.

**Figure supplement 4.** The responsible sequence of the *bd* mutant.

**Figure supplement 4—source data 1.** The genomic sequence of the *Bm-mamo* gene and its upstream region in the wild-type and *bd* mutant.

*Figure 2 continued on next page*

*Figure 2 continued*
**Figure supplement 5.** Sequence alignment between the mamo protein of *Drosophila melanogaster* and the amino acid sequence encoded by the *012517/18* gene.

**Figure supplement 5—source data 1.** The amino acid sequences encoded of the *mamo* genes in silkworms and *Drosophila melanogaster*.

*supplement 5*). Therefore, we named the *12517/18* gene *Bm-mamo*; the long transcript was designated *Bm-mamo-L*, and the short transcript was designated *Bm-mamo-S*.

The *mamo* gene belongs to the Broad-complex, Tramtrack and bric à brac/poxvirus zinc finger protein (BTB-ZF) family. In the BTB-ZF family, the zinc finger serves as a recognition motif for DNA-specific sequences, and the BTB domain promotes oligomerization and the recruitment of other regulatory factors (*Bardwell and Treisman, 1994*). Most of these factors are transcriptional repressors, such as nuclear receptor corepressor (N-CoR) and silencing mediator for retinoid and thyroid hormone receptor (SMRT) (*Huynh and Bardwell, 1998*), but some are activators, such as p300 (*Staller et al., 2001*). Therefore, these features commonly serve as regulators of gene expression. *mamo* is enriched in embryonic primordial germ cells in *D. melanogaster*. Individuals deficient in *mamo* are able to undergo oogenesis but fail to execute meiosis properly, leading to female infertility in *D. melanogaster* (*Mukai et al., 2007*). *Bm-mamo* was identified as an important candidate gene for further analysis.

## Expression pattern analysis of *Bm-mamo*

To analyze the expression profiles of *Bm-mamo*, we performed quantitative PCR. The expression levels of the *Bm-mamo* gene in the Dazao strain were investigated throughout the body at different developmental stages, from the embryonic stage to the adult stage. The gene was highly expressed in the molting stage of caterpillars, and its expression was upregulated in the later pupal and adult stages (*Figure 3A*). This finding suggested that the *Bm-mamo* gene responds to ecdysone and participates in the processes of molting and metamorphosis in silkworms. According to the investigation of tissue-specific Dazao strain expression in 5th-instar 3rd-day larvae, the midgut, head, and epidermis exhibited high expression levels; the trachea, nerves, silk glands, testis, ovary, muscle, wing disc, and Malpighian tubules exhibited moderate expression levels; and the blood and fat bodies exhibited low expression levels (*Figure 3B*). These findings suggested that *Bm-mamo* is involved in the regulation of the development of multiple silkworm tissues. Due to the melanism of the epidermis of the *bd* mutant and the high expression level of the *Bm-mamo* gene in the epidermis, we measured the expression level of this gene in the epidermis of the 4th to 5th instars of the Dazao strain. In the epidermis, the *Bm-mamo* gene was upregulated during the molting period, and the highest expression was observed at the beginning of molting (*Figure 3C*).

## Functional analyses of *Bm-mamo*

To study the function of the *Bm-mamo* gene, we carried out an RNA interference (RNAi) experiment. Short interfering RNA (siRNA) was injected into the hemolymph of silkworms, and an electroporation experiment (*Ando and Fujiwara, 2013*) was immediately conducted. We found significant melanin pigmentation in the epidermis of the newly molted 5th-instar larvae. These findings indicate that *Bm-mamo* deficiency can cause melanin pigmentation (*Figure 4*). The melanistic phenotype of the RNAi individuals was similar to that of the *bd* mutants.

In addition, gene knockout was performed. Ribonucleoproteins (RNPs) generated from guide RNA (gRNA) and recombinant Cas9 protein were injected into 450 silkworm embryos. In the G0 generation, individuals with a mosaic melanization phenotype were found. These melanistic individuals were raised to moths and subsequently crossed. A homozygous line with a gene knockout was obtained through generations of screening. The gene-edited individuals had a significantly melanistic body color, and the female moths were sterile (*Figure 5*).

These results indicated that the *Bm-mamo* gene negatively regulates melanin pigmentation in caterpillars and participates in the reproductive regulation of female moths.

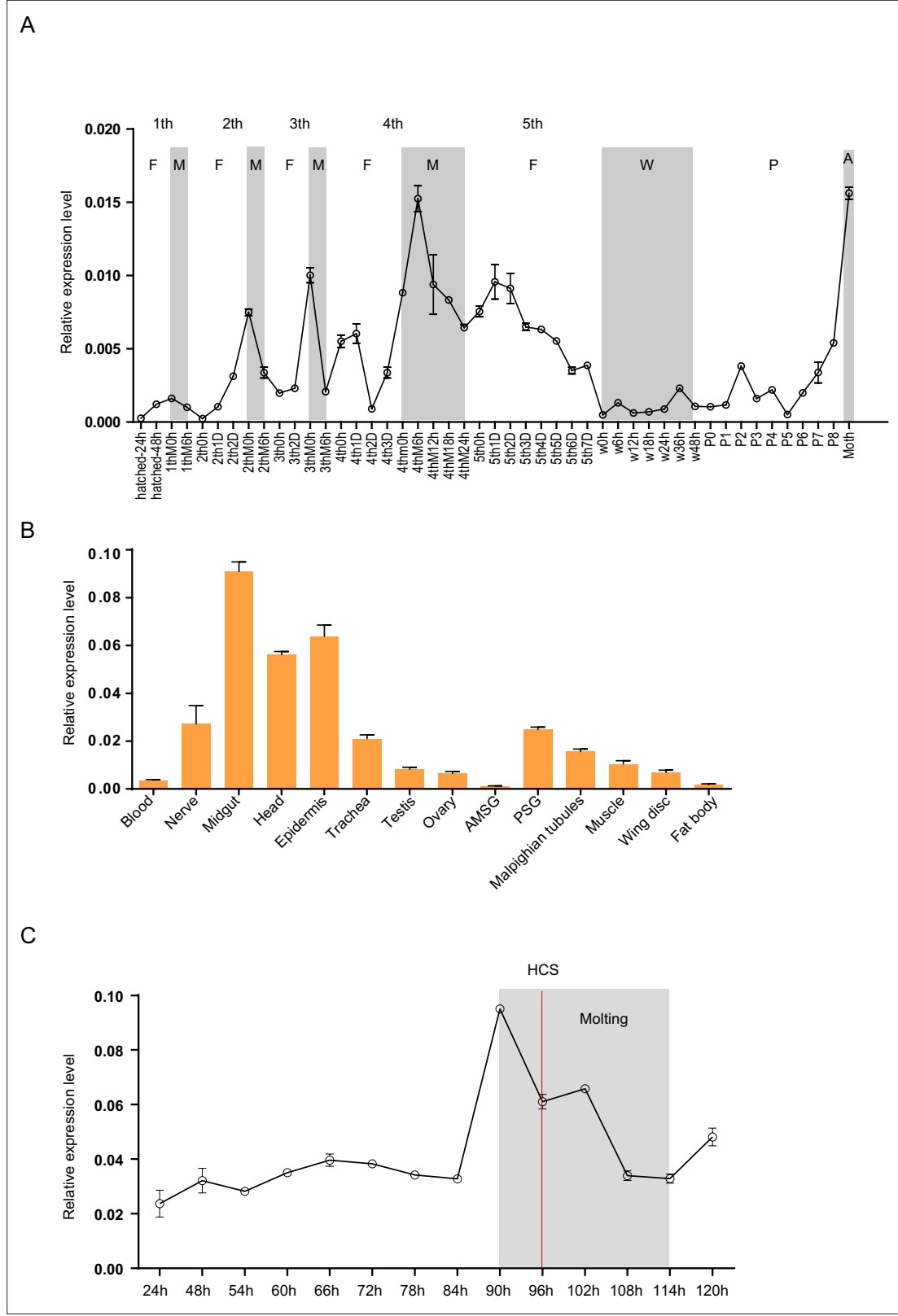

**Figure 3.** Spatiotemporal expression of *Bm-mamo*. (**A**) Temporal expression of *Bm-mamo*. In the molting and adult stages, this gene was significantly upregulated. M: molting stage, W: wandering stage, P: pupal stage, P1: Day 1 of the pupal stage, A: adult stage, 4th3d indicates the 4th instar on Day 3. 1st to 5th denote the first instar of larvae to fifth instar of larvae, respectively. (**B**) Tissue-specific expression of genes in 4th-instar molting larvae. *Bm-mamo* had relatively high expression levels in the midgut, head, and epidermis. AMSG: anterior division of silk gland and middle silk gland, PSG:

*Figure 3 continued on next page*

*Figure 3 continued*

posterior silk gland. (**C**) Detailed analysis of *Bm-mamo* at the 4th larval stage in the epidermis of the Dazao strain. *Bm-mamo* expression is upregulated during the molting stage. HCS indicates the head capsule stage. The 'h' indicates the hour, 90 h: at 90 h of the 4th instar. The mean ± SD, n=3.

## Downstream target gene analysis

*Bm-mamo* belongs to the zinc finger protein family, which specifically recognizes downstream DNA sequences according to their zinc fingers. We identified homologous genes of *mamo* in multiple species and conducted a phylogenetic analysis (*Figure 6*). Because the sequences of zinc finger motifs are key for downstream target gene recognition, we compared the zinc finger motifs of these orthologous proteins. Sequence alignment revealed that the amino acid residues of the zinc finger motif of the mamo-S protein were highly conserved among 57 species. This may also be due to incomplete prediction of alternative splicing in GenBank, resulting in fewer homologous proteins in mamo-L than in mamo-S (*Figure 6—figure supplement 1*). As the mamo protein contains a tandem $Cys_2His_2$ zinc finger (C2H2-ZF) motif, it can directly bind to DNA sequences. Previous research has suggested that the ZF-DNA-binding interface can be understood as a 'canonical binding model', in which each finger contacts DNA in an antiparallel manner. The binding sequence of the C2H2-ZF motif is determined by the amino acid residue sequence of its α-helical component. The first amino acid residue in the α-helical region of the C2H2-ZF domain is at position 1, and positions −1, 2, 3, and 6 are key amino acids involved in the recognition and binding of DNA. The residues at positions −1, 3, and 6 specifically interact with base 3, base 2, and base 1 of the DNA sense sequence, respectively, while the residue at position 2 interacts with the complementary DNA strand (*Pabo et al., 2001*; *Wolfe et al., 2000*). To analyze the downstream target genes of mamo, we first predicted the DNA-binding motifs of these genes using online software (http://zf.princeton.edu) based on the canonical binding model (*Persikov and Singh, 2014*). In addition, the DNA-binding sequence of mamo (TGCGT) in *Drosophila* was confirmed by electrophoretic mobility shift assay (EMSA) (*Hira et al., 2013*), which has a consensus sequence with the predicted binding site sequence of Bm-mamo-S (GTGCGTGGC), and the predicted sequence was longer. This indicates that the predicted results for the DNA-binding site have good reliability. Furthermore, the predicted DNA-binding sites of Bm-mamo-L and Bm-mamo-S were highly consistent with those of mamo orthologs in different species (*Figure 6—figure supplement 2*). This finding suggested that the protein may regulate similar target genes between species.

C2H2-ZF transcription factors function by recognizing and binding to *cis*-regulatory sequences in the genome, which harbor *cis*-regulatory elements (CREs) (*Wittkopp and Kalay, 2012*). CREs are broadly classified as promoters, enhancers, silencers, or insulators (*Preissl et al., 2023*). CREs are often near their target genes, such as enhancers, which are typically located upstream (5′) or downstream (3′) of the gene they regulate or in introns, but approximately 12% of CREs are located far from their target gene (*Kvon et al., 2014*). Therefore, we first investigated the 2 kb upstream and downstream regions of the predicted genes in silkworms.

The predicted position weight matrices (PWMs) of the recognized sequences of Bm-mamo protein and the Find Individual Motif Occurrences (FIMO) software of MEME were used to perform silkworm whole-genome scanning for possible downstream target genes. The 2 kb upstream and downstream regions of the 14,623 predicted genes in silkworms were investigated. A total of 10,622 genes contained recognition sites within 2 kb of the upstream/downstream region of the Bm-mamo protein in the silkworm genome (*Figure 7*, *Supplementary file 3–Table S3*, and *Supplementary file 4–Table S4*).

Moreover, we compared the transcriptome data of integument tissue between homozygotes and heterozygotes of the *bd* mutant at the 4th instar/beginning molting stage (*Wu et al., 2016*). In the integument tissue, 10,072 genes (~69% of the total predicted genes of silkworm) were expressed in heterozygotes, and 9853 genes (~67% of the total predicted genes) were expressed in homozygotes of the *bd* mutant. In addition, there were 191 genes whose expression significantly differed between homozygotes (*bd/bd*) and heterozygotes (+/*bd*) according to comparative transcriptome analysis (*Supplementary file 5–Table S5*; *Wu et al., 2016*). Protein functional annotation was performed, and 19 CP genes were significantly differentially expressed between heterozygotes and homozygotes of *bd*. In addition, the orthologs of these CPs were analyzed in *Danaus plexippus*, *Papilio xuthu*, and *D. melanogaster* (*Supplementary file 6–Table S6*). Furthermore, we identified 53 enzyme-encoding genes, 17 antimicrobial peptide genes, 6 transporter genes, 5 transcription factor genes, 5 cytochrome

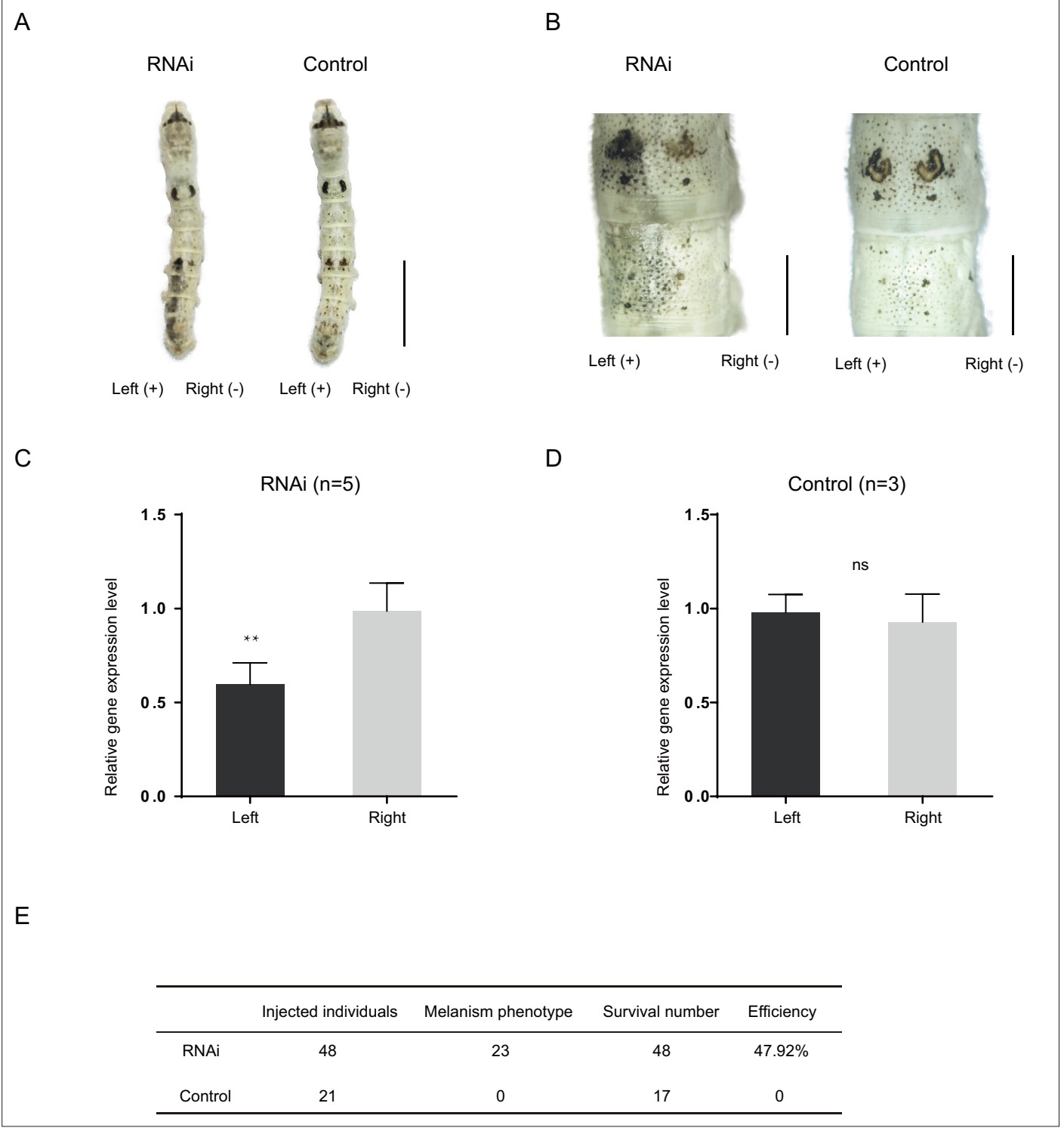

**Figure 4.** RNA interference (RNAi) of *Bm-mamo*. (**A**) Short interfering RNA (siRNA) was introduced by microinjection followed by electroporation. '+' and '−' indicate the positive and negative poles of the electrical current, respectively. Scale bars, 1 cm. (**B**) Partial magnification of the siRNA experimental group and negative control group. Scale bars, 0.2 cm. (**C**, **D**) Relative expression levels of *Bm-mamo* in the negative control and RNAi groups were determined by quantitative polymerase chain reaction (qPCR) analysis. The mean ± SD. Numbers of samples are shown in the upper right in each graph. **$p < 0.01$, paired Student's *t* test (ns, not significant). (**E**) Statistical analysis of the efficiency of the RNAi.

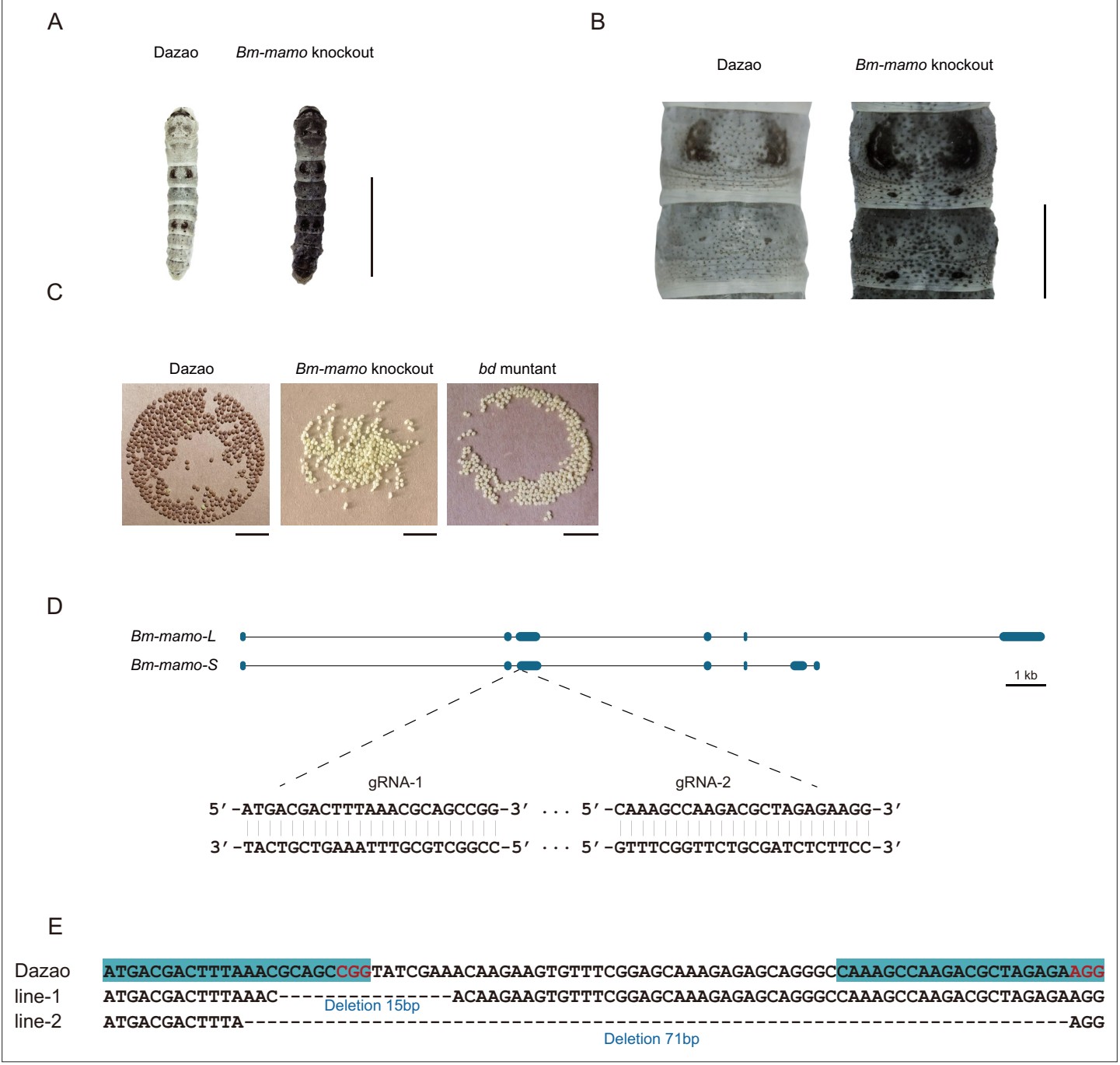

**Figure 5.** *Bm-mamo* knockout. (**A**) Larval phenotype of G3 fourth-instar larvae of *Bm-mamo*-targeted Dazao. Scale bars, 1 cm. (**B**) Partial magnification of *Bm-mamo* knockout and control individuals. Scale bars, 0.2 cm. (**C**) After 48 hr of egg laying, the pigmentation of the eggs indicates that those produced by homozygous knockout females cannot undergo normal pigmentation and development. (**D**) Genomic structure of *Bm-mamo*. The open reading frame (blue) and untranslated region (black) are shown. The gRNA 1 and gRNA 2 sequences are shown. (**E**) Sequences of the *Bm-mamo* knockout individuals. Lines 1 and 2 indicate deletions of 15 and 71 bp, respectively.

genes, and others. Among the differentially expressed genes (DEGs), CPs were significantly enriched, and previous studies have shown that CPs can affect pigmentation (*Xiong et al., 2017*). Therefore, we first investigated the expression of the CP genes. Among them, 18 CP genes had Bm-mamo binding sites in the upstream and downstream 2 kb genomic regions. In addition, we investigated the expression levels of the 18 CP genes in the integument from the 4th instar (Day 1) to the beginning of the 5th instar in the *Bm-mamo* knockout lines (*Figure 8*). Compared with those in heterozygous (*mamo⁻/+*)

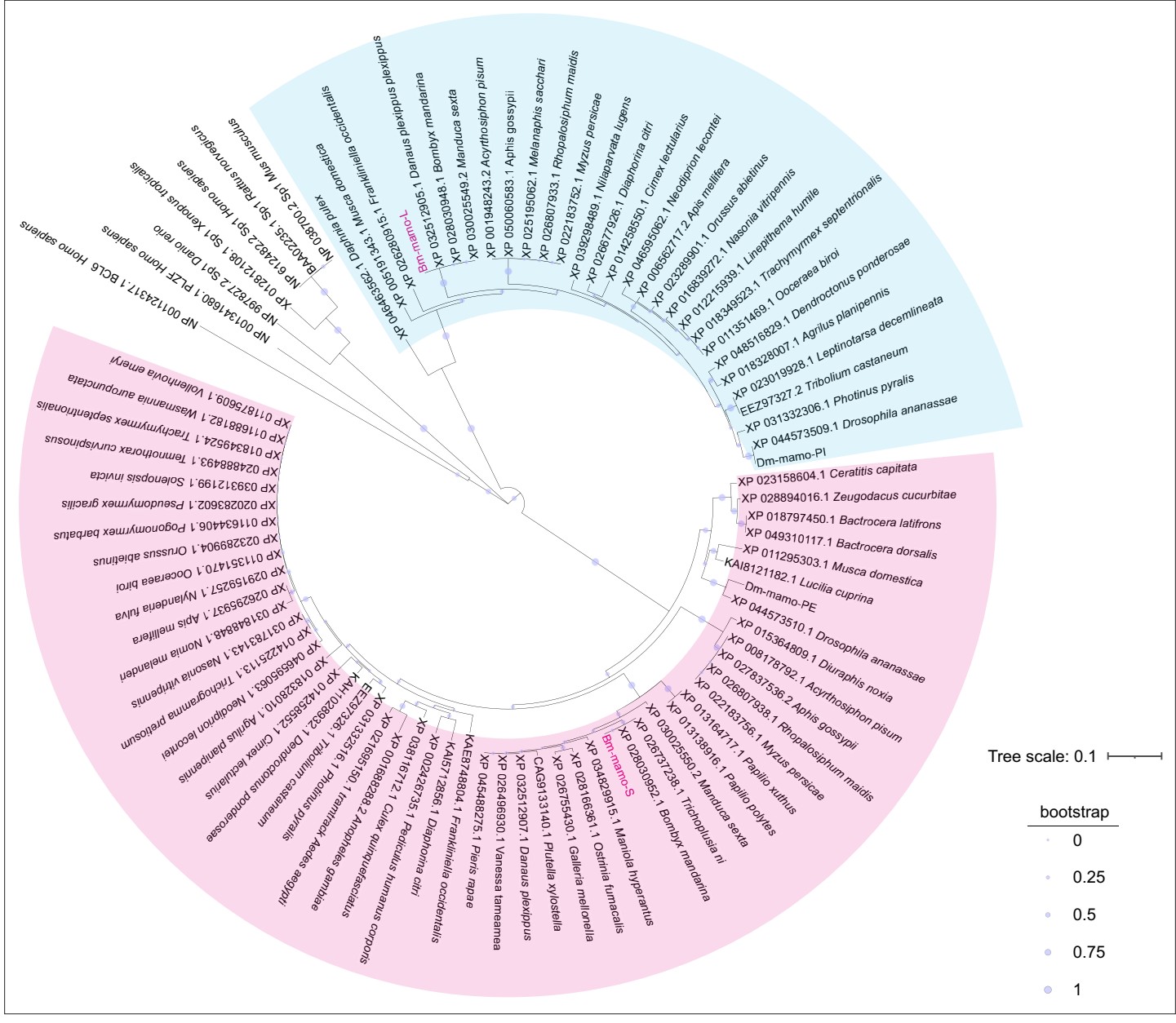

**Figure 6.** Phylogenetic analysis. The phylogenetic tree was subsequently constructed using the neighbor-joining method with the MEGA7 program (Pearson model). The confidence levels for various phylogenetic lineages were estimated by bootstrap analysis (2000 replicates). The magenta color indicates the homologous Bm-mamo protein, which has three zinc finger motifs. The blue color indicates the homologous Bm-mamo protein, which has five zinc finger motifs.

The online version of this article includes the following source data and figure supplement(s) for figure 6:

**Source data 1.** The amino acid sequences of the mamo orthologs among multiple species.

**Figure supplement 1.** The sequence alignment of the zinc finger motif of orthologs of mamo in multiple species.

**Figure supplement 1—source data 1.** The amino acid sequences of the zinc finger motifs of the mamo orthologs among multiple species.

**Figure supplement 2.** The sequence logo of the DNA-binding sites of orthologs of mamo in multiple species.

*Bm-mamo* knockout individuals, CP genes were significantly upregulated at one or several time points in homozygous (*mamo⁻/mamo⁻*) individuals. Interestingly, the expression of the CP gene *BmorCPH24* was significantly upregulated at the feeding stage in *Bm-mamo* knockout homozygotes. Previous studies have shown that BmorCPH24 deficiency can lead to a marked decrease in pigmentation in

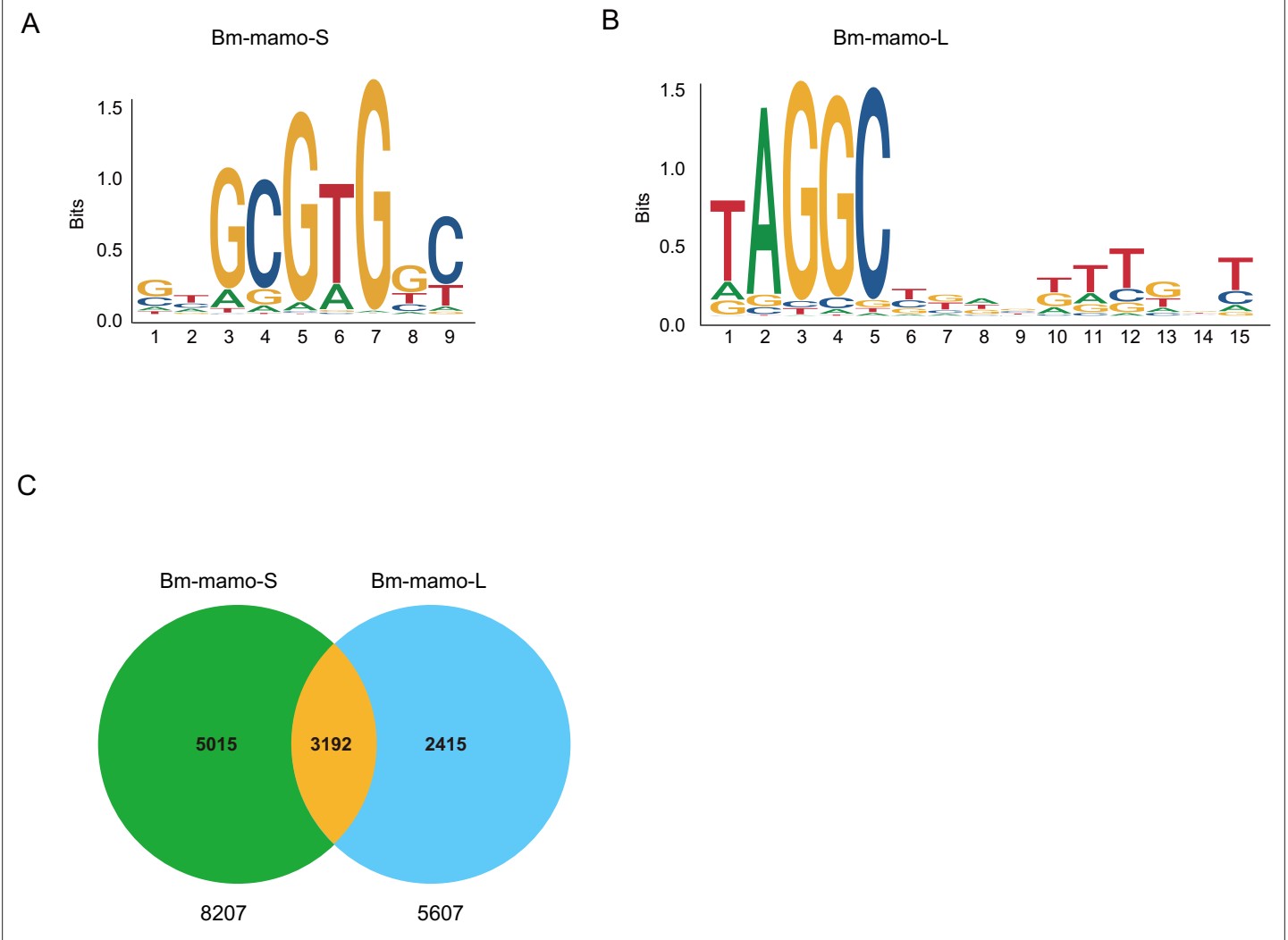

**Figure 7.** The DNA-binding site sequence logo of the Bm-mamo protein and the analysis of downstream target genes. (**A**) The DNA-binding site sequence logo of the Bm-mamo-S protein. (**B**) The DNA-binding site sequence logo of the Bm-mamo-L protein. (**C**) Potential downstream target genes of the Bm-mamo protein in the silkworm genome were identified via the MEME FIMO program.

silkworm larvae (*Xiong et al., 2017*). Therefore, the expression of some CP genes may be necessary for determining the color patterns of caterpillars.

In addition, the synthesis of pigment substances is an important driver of color patterns (*Wittkopp et al., 2003a*). Eight key genes (*TH*, *DDC*, *aaNAT*, *ebony*, *black*, *tan*, *yellow*, and *laccase2*) involved in melanin synthesis (*Matsuoka and Monteiro, 2018*) were investigated in the heterozygous and homozygous *Bm-mamo* gene knockout lines. Among them, *yellow*, *tan*, and *DDC* were significantly upregulated during the molting period in the homozygous *Bm-mamo* knockout individuals (*Figure 9*). The upstream and/or downstream 2 kb genomic regions of *yellow*, *tan*, and *DDC* contain the binding site of the Bm-mamo protein. In addition, the expression of *yellow*, *DDC*, and *tan* can promote the generation of melanin (*Gibert, 2020*).

To explore the interaction between the Bm-mamo protein and its binding sequence, EMSA was conducted. The binding site of Bm-mamo-S (CTGCGTGGT) was located approximately 70 bp upstream of the transcription initiation site of the *Bm-yellow* gene. The EMSA results showed that the Bm-mamo-S protein expressed in prokaryotes can bind to the CTGCGTGGT sequence in vitro (*Figure 9—figure supplement 1*). This finding suggested that the Bm-mamo-S protein can bind to the upstream region of the *Bm-yellow* gene and regulate its transcription. Therefore, the *Bm-mamo*

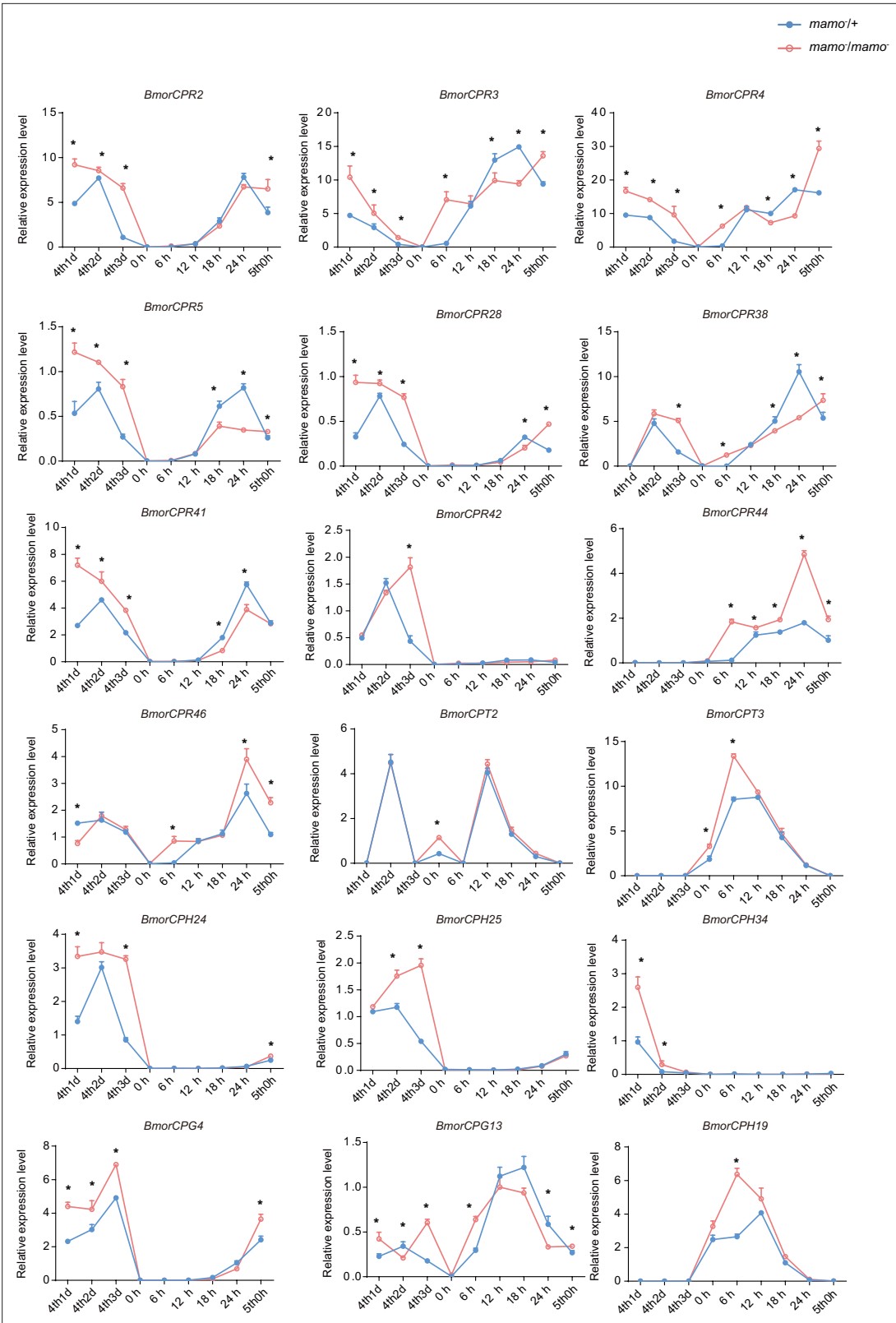

**Figure 8.** The expression levels of 18 cuticular protein genes were measured via quantitative polymerase chain reaction (qPCR). The red line indicates the homozygous *Bm-mamo* knockout individuals, and the blue line indicates the heterozygous *Bm-mamo* knockout individuals. The mean ± SD, n=3. *$p$ < 0.05 ( Student's $t$ test). 4th1d indicates 4-instar Day 1; 0 h indicates the beginning of molting, h indicates hours, and 5th0h indicates the beginning of the 5th instar.

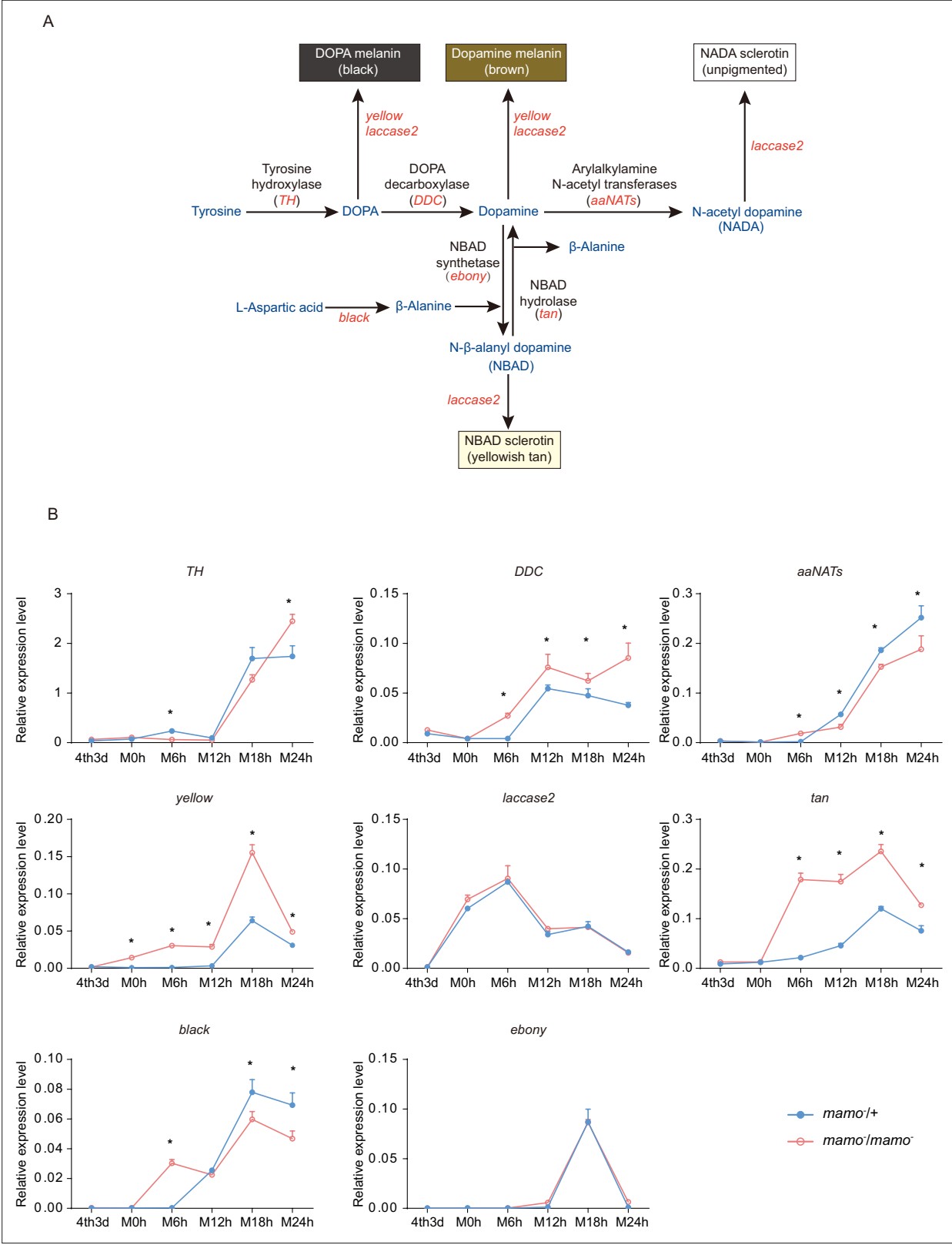

**Figure 9.** Melanin metabolism pathway and quantitative polymerase chain reaction (qPCR) of related genes. (**A**) The melanin metabolism pathway. Blue indicates amino acids and catecholamines, red indicates the names of genes, and black indicates the names of enzymes. (**B**) Relative expression levels of eight genes in the heterozygous *Bm-mamo* knockout group (blue) and homozygous *Bm-mamo* knockout group (red) determined via qPCR analysis. The mean ± SD, n=3. *$p < 0.05$ (Student's *t* test). 4th3d indicates the 4th instar on Day 3, M indicates molting, and h indicates hours.

*Figure 9 continued on next page*

**Figure supplement 1.** Bm-mamo directly binds *yellow*.

**Figure supplement 3.** The *cis*-regulatory element model of *yellow* in *D. melanogaster*.

**Figure supplement 3—source data 1.** Genomic sequence of *yellow* gene in *Drosophila melanogaster*.

gene may control the color pattern of caterpillars by regulating key melanin synthesis genes and 18 CP genes.

## Discussion

Insects have evolved many important phenotypes during the process of adapting to the environment. Among these traits, the color pattern is one of the most interesting. The biochemical metabolic pathways of pigments, such as melanin, ommochromes, and pteridines, have been identified in insects (*Tong et al., 2021*). However, the regulation of pigment metabolism-related genes and the processes involved in the transport and deposition of pigment substances are unclear. In this study, we discovered that the *Bm-mamo* gene negatively regulates melanin pigmentation in caterpillars. When this gene is deficient, the body color of silkworms exhibits substantial melanism, and changes in the expression of some melanin synthesis genes and some CP genes are also significant.

### Structural genes involved in melanin synthesis

The deficiencies of some genes in the melanin synthesis pathway, such as *TH*, *yellow*, *ebony*, *tan*, and *aaNAT*, can lead to variations in color patterns (*Matsuoka and Monteiro, 2018*). However, they often lead to localized pigmentation changes in later-instar caterpillars. For example, in *yellow*-mutant silkworms, the eye spot, lunar spot, star spot, spiracle plate, head, and some sclerotized areas appear reddish brown, and the other cuticle is consistent with that in the wild type in later instars of larvae (third to fifth instars) (*Futahashi et al., 2008b*). This situation is highly similar to that of the *tan* mutant in silkworms (*Noh et al., 2016*). Why do the *yellow* and *tan* phenotypes appear only in a limited cuticle region in later instars of silkworm larvae? One possible reason is that the expression of *yellow* and *tan* is limited to a certain region by transcription regulators. Alternatively, other factors, such as CPs in the cuticle, may limit pigmentation by interacting with pigment substances in the cuticle.

On the one hand, we investigated the expression levels of *yellow* and *tan* in the pigmented region (lunar spot) and nonpigmented region of the epidermis during the 4th molting of the wild-type Dazao

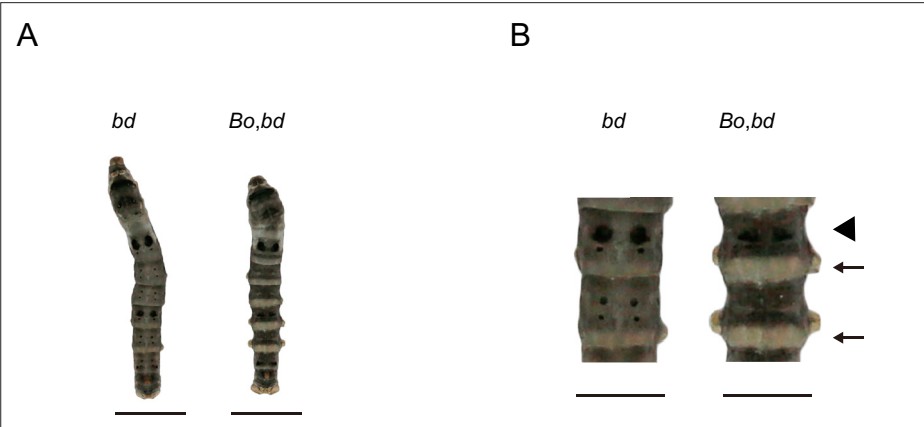

**Figure 10.** Phenotypes of offspring from hybridization between *Bo* and *bd*. (**A**) The phenotypes of 5th instar and Day 3. the. The bars indicate 1 cm. (**B**) Magnification of body segments. The bars indicate 0.5 cm. The area where the pigment is significantly reduced is indicated by the arrow. The triangle shape indicates that the number of lunar spots is reduced.

strain. The expression level of *yellow* was significantly upregulated in the lunar spot (the epidermis on the dorsum of the fifth body segment) compared with the nonpigmented region (the epidermis on the dorsum of the sixth body segment) at 6 and 12 hr after molting. Moreover, *tan* was significantly upregulated 18 hr after molting in the lunar spot region (*Figure 9—figure supplement 2*). It is suggested that the upregulated expression of pigment synthesis genes at key time points may be important for pigmentation. However, *yellow* and *tan* were still moderately expressed in the nonpigmented epidermis, although they did not cause significant melanin pigmentation. This finding indicates that pigment synthesis alone cannot determine the predominant color pattern of the cuticle in caterpillars.

## CPs participate in pigmentation

On the other hand, synthesized pigment substances need to be transported from epidermal cells and embedded into the cuticle to allow pigmentation. Therefore, the correct cuticle structure and location of CPs in the cuticle may be important factors affecting pigmentation. Previous studies have shown that a lack of *BmorCPH24*, which encodes an important component of the endocuticle, can lead to dramatic changes in body shape and a significant reduction in the pigmentation of caterpillars (*Xiong et al., 2017*). We crossed *Bo* (*BmorCPH24* null mutation) and *bd* to obtain $F_1$ (*Bo/+*$^{Bo}$, *bd/+*) and then self-crossed $F_1$ to observe the phenotype of $F_2$. The area of lunar spots and star spots decreased, and light-colored stripes appeared on the body segments; however, the other areas still exhibited significant melanin pigmentation in double-mutation (*Bo*, *bd*) individuals (*Figure 10*). However, in previous studies, the introduction of *Bo* into *L* (the ectopic expression of *wnt1* results in lunar stripes generated on each body segment) (*Yamaguchi et al., 2013*) and *U* (the overexpression of *SoxD* results in excessive melanin pigmentation of the epidermis) (*Wang et al., 2022*) strains by genetic crosses markedly reduced the pigmentation of *L* and *U* (*Xiong et al., 2017*). Interestingly, there was a more significant decrease in pigmentation in the double mutants (*Bo*, *L*) and (*Bo*, *U*) than in (*Bo*, *bd*). These findings suggested that *Bm-mamo* has a greater ability than does *wnt1* and *SoxD* to regulate pigmentation. On the one hand, mamo may be a stronger regulator of the melanin metabolic pathway, and on the other hand, mamo may regulate other CP genes to reduce the impact of BmorCPH24 deficiency.

How do CPs affect pigmentation? One study showed that some CPs can form 'pore canals' to transport macromolecules (*Kramer et al., 2001*). In addition, some CPs can be crosslinked with catecholamines, which are synthesized via the melanin metabolism pathway (*Noh et al., 2015*). Because there are no live cells in the cuticle, melanin precursor substances may be transported by the pore canals formed by some CPs and fixed to specific positions through cross-linking with CPs. The CP TcCPR4 is needed for the formation of pore canals in the cuticle of *Tribolium castaneum* (*Noh et al., 2014*). In contrast, the vertical pore canal is lacking in the less pigmented cuticles of *T. castaneum* (*Mun et al., 2015*). This finding suggested that the pore canals constructed by TcCPR4 may transport pigments and contribute to cuticle pigmentation in *T. castaneum*. Moreover, the melanin metabolites *N*-acetyldopamine and *N*-β-alanyldopamine can target and sclerotize the cuticle by cross-linking with specific CPs (*Noh et al., 2016*). This finding suggested that pigments interact with specific CPs, thereby affecting pigmentation, hardening properties, and the structure of the cuticle. Interestingly, a study showed that in addition to absorbing specific wavelengths, pigments can affect cuticle polymerization, density, and the refractive index, which in turn affects the reflected wavelengths that produce structural color in butterfly wing scales (*Prakash et al., 2023*). This implies that the interaction between pigments and CPs can be very subtle, resulting in the formation of unique nanostructures, such as those on wing scales, that produce brilliant structural colors.

Consequently, to maintain the accuracy of the color pattern, the localization of CPs in the cuticle may be very important. In a previous study employing microarray analysis, different CPs were found in differently colored areas of the epidermis in *P. xuthus* larvae (*Futahashi et al., 2012*). We investigated whether the CP genes were highly expressed in the black region of *P. xuthus* caterpillars. Thirteen orthologous genes were found in silkworms (*Supplementary file 7–Table S7*). Among them, 11 genes (*BmorCPR67*, *BmorCPR71*, *BmorCPR76*, *BmorCPR79*, *BmorCPR99*, *BmorCPR107*, *BmorCPT4*, *BmorCPH5*, *BmorCPFL4*, *BmorCPG27*, and *BmorCPG4*) were significantly upregulated in homozygous (*mamo⁻/mamo⁻*) knockout individuals at some time points from the 3rd day of the 4th instar to the 5th instar; *BmorCPG4* was also among the 18 previously detected CP genes, and two genes (*BmorCPG38* and *BmorCPR129*) were not differentially expressed between homozygous (*mamo⁻/mamo⁻*) and heterozygous (*mamo⁻/+*) individuals (*Figure 11*).

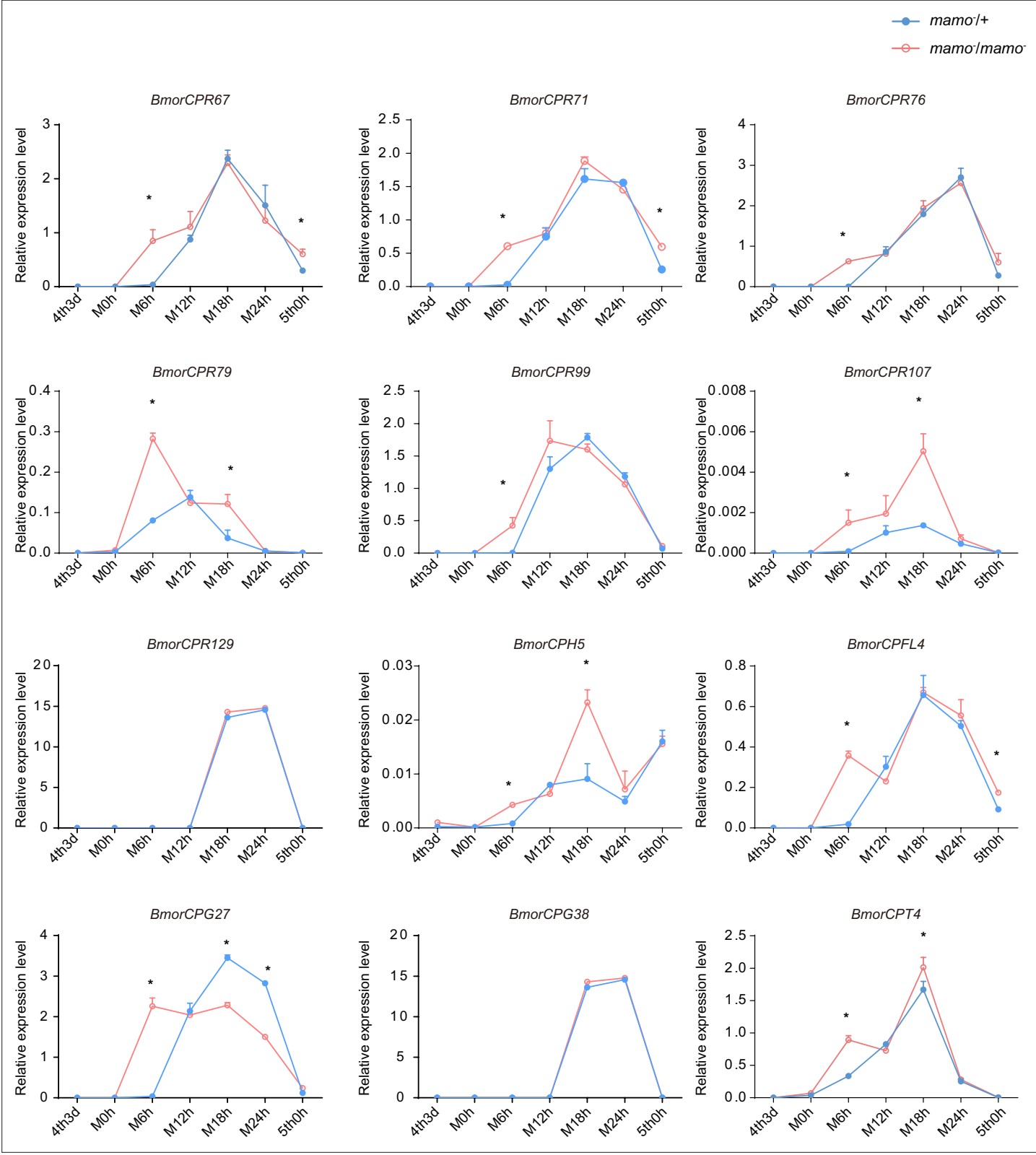

**Figure 11.** The expression levels of 12 cuticular protein-encoding genes were detected. The expression levels of the orthologous genes of the cuticular protein genes highly expressed in the black marking of *Papilio xuthus* caterpillars in the silkworm caterpillars were detected. Blue indicates heterozygous *Bm-mamo* gene knockout individuals, and red indicates homozygous *Bm-mamo* gene knockout individuals. The mean ± SD, n=3. *$p < 0.05$ (Student's *t* test). 4th3d indicates the 4th instar on Day 3, M indicates molting, and h indicates hours.

*Figure 11 continued on next page*

*Figure 11 continued*

The online version of this article includes the following figure supplement(s) for figure 11:

**Figure supplement 1.** Heatmap of 28 differentially expressed cuticular protein genes.

The expression of the 28 CP genes mentioned above was significantly upregulated in homozygous ($mamo^-/mamo^-$) gene knockout individuals at several stages. These CPs may be involved in the transport or cross-linking of melanin in the cuticle. However, there were no differences in the expression levels of these genes during some periods compared with those in the control group, and the expression of some genes was significantly downregulated at some time points in the melanic individuals (*Figure 11—figure supplement 1*). This finding suggested that the regulation of CP genes is complex and may involve other transcription factors and feedback effects. CPs are essential components of the insect cuticle and are involved in cuticular microstructure construction (*Noh et al., 2017*), body shape development (*Tajiri et al., 2017*), wing morphogenesis (*Zhao et al., 2019*), and pigmentation (*Xiong et al., 2017*). CP genes usually account for more than 1% of the total genes in an insect genome and can be categorized into several families, including CPR, CPG, CPH, CPAP1, CPAP3, CPT, CPF, and CPFL (*Willis, 2010*). The CPR family is the largest group of CPs and contains a chitin-binding domain called the Rebers and Riddiford motif (R&R) (*Rebers and Riddiford, 1988*). The R&R consensus sequences can be divided into three subfamilies (RR-1, RR-2, and RR-3) (*Karouzou et al., 2007*). Among the 28 CPs, 11 RR-1 genes, 6 RR-2 genes, 4 hypothetical cuticular protein (CPH) genes, 3 glycine-rich cuticular protein (CPG) genes, 3 cuticular protein Tweedle motif (CPT) genes, and 1 CPFL (like the CPFs in a conserved C-terminal region) gene were identified. The RR-1 consensus among species is usually more variable than the RR-2 consensus, which suggests that RR-1 may have a species-specific function. RR-2 often clustered into several branches, which may be due to gene duplication events in co-orthologous groups and may result in conserved functions between species (*Chen and Hou, 2021*). The classification of CPH was based on the lack of known motifs. In the epidermis of Lepidoptera, CPH genes often exhibit high expression levels. For example, *BmorCPH24* has the highest expression level in the silkworm larval epidermis (*Fu et al., 2023*). The CPG protein is rich in glycine. The CPH and CPG genes are less commonly found in insects outside the order Lepidoptera (*Yang et al., 2017*). This finding suggested that these genes may provide species-specific functions for Lepidoptera. CPT contains a Tweedle motif, and the *TweedleD1* mutation has a dramatic effect on body shape in *D. melanogaster* (*Guan et al., 2006*). The CPFL family members are relatively conserved among species and may be involved in the synthesis of larval cuticles (*Togawa et al., 2007*). CPT and CPFL may have relatively conserved functions among insects. The CP genes are a group of rapidly evolving genes, and their copy numbers may undergo significant changes in different species. In addition, RNAi experiments on 135 CP genes in the brown planthopper (*Nilaparvata lugens*) revealed that deficiency of 32 CP genes leads to significant defects in phenotypes, such as lethality and developmental retardation. These findings suggested that the 32 CP genes are indispensable and that other CP genes may have redundant and complementary functions (*Pan et al., 2018*). Previous studies revealed that the construction of the larval cuticle of silkworms requires the precise expression of more than two hundred CP genes (*Yan et al., 2022*). The production, interaction, and deposition of CPs and pigments are complex and precise processes, and our research showed that *Bm-mamo* plays an important regulatory role in this process in silkworm caterpillars. To further understand the role of CPs, future work should aim to identify the functions of important CP genes and the deposition mechanism in the cuticle.

## Maturation of pigment granules

In addition, among the 191 DEGs found in the comparative transcriptome data, we also discovered several interesting genes. For example, *BGIBMGA013242*, which encodes a major facilitator superfamily protein (MFS) named *BmMFS* and is responsible for the *cheek and tail spot (cts)* mutant, was significantly upregulated in the *bd* mutant. A deficiency of this gene (*BmMFS*) results in chocolate-colored head and anal plates on the silkworm caterpillar (*Ito et al., 2012*). In the ommochrome metabolic pathway, *Bm-re*, which encodes an MFS protein, may function in the transportation of several amino acids, such as cysteine or methionine, into pigment granules (*Osanai-Futahashi et al., 2012*; *Luo et al., 2021*). Therefore, the encoded product of *BmMFS* may participate in pigmentation by promoting the maturation of pigment granules. Moreover, *BGIBMGA013576* and *BGIBMGA013656*, which encode MFS domain-containing proteins belonging to the solute carrier family 22 (SLC22) and

solute carrier family 2 (SLC2) families, respectively, were significantly upregulated in *bd*. These two genes may participate in pigmentation in a similar manner to that of the *BmMFS* gene. In addition, the *red Malpighian tubule* (*red*) gene encodes a LysM domain-containing protein. *Red* deficiency results in a significant decrease in orange pterin in the wing of *Colias* butterflies. Research suggests that the product of *red* may interact with V-ATPase to modulate vacuolar pH across a variety of endosomal organelles, thereby affecting the maturation of pigment granules in cells (*Hanly et al., 2023*). These findings indicate that the maturation of pigment granules plays an important role in the coloring of Lepidoptera and that *Bm-mamo* may be involved in regulating the maturation of pigment granules in the epidermal cells of silkworms.

Bm-mamo may affect the synthesis of melanin in epidermal cells by regulating *yellow*, *DDC*, and *tan*, regulating the maturation of melanin granules in epidermal cells through *BmMFS*, and affecting the deposition of melanin granules in the cuticle by regulating CP genes, thereby comprehensively regulating the color pattern of caterpillars.

## The upstream sequence of *Bm-mamo*

Moreover, in *D. melanogaster*, *mamo* is needed for functional gamete production. In the silkworm, *Bm-mamo* has a conserved function in female reproduction. The DNA-binding motifs of the mamo proteins in silkworms and fruit flies are highly conserved, suggesting that they may regulate the same downstream target genes. However, *mamo* has developed new functions in color pattern regulation in silkworm caterpillars. We found that in *D. melanogaster*, the *mamo* gene is expressed mainly during the adult stage; it is not expressed during the 1st instar or 2nd instar larval stage and has very low expression at the 3rd instar (*Supplementary file 8–Table S8*). In addition, several binding sites of mamo were found near the TSS of *yellow* in *D. melanogaster* (*Figure 9—figure supplement 3*). The *yellow* gene is a key melanin metabolic gene with upstream and intron sequences that have been identified as multiple CREs, and it has been considered a research model for CREs (*Kalay et al., 2019*; *Xin et al., 2020*). This difference may be due to a change in the expression pattern of *mamo*, which led to the development of a new function for this gene in regulating coloration in silkworm caterpillars.

Changes in gene expression patterns are generally believed to be the result of the evolution of CREs. Because CREs play important roles in the spatiotemporal expression pattern regulation of genes, many CREs are found in noncoding regions, which are relatively prone to sequence variation compared with the sequences of coding genes (*Wittkopp and Kalay, 2012*). However, the molecular mechanism underlying the sequence changes in CREs is unclear.

Transcription factors (TFs), because they recognize relatively short sequences, generally between 4 and 20 bp in length, can have many binding sites in the genome (*Jolma and Taipale, 2011*). Therefore, one member of the TF family has the potential to regulate many genes in one genome. Single regulation of specific TFs can lead to patterned changes in the expression of multiple downstream genes, enabling organisms to adapt to the environment by altering a given type of trait. For example, the marine form of the three-spined stickleback (*Gasterosteus aculeatus*) has thick armor, whereas the lake population (which was recently derived from the marine form) does not. Research has shown that pelvic loss in different natural populations of three-spined stickleback fishes occurs via regulatory mutations resulting from the deletion of a tissue-specific enhancer (*Pel*) of the *pituitary homeobox transcription factor 1* (*Pitx1*) gene. The researchers genotyped 13 pelvic-reduced populations of three-spined sticklebacks from disparate geographic locations. Nine of the 13 pelvic-reduced stickleback populations had sequence deletions of varying lengths, all of which were located at the *Pel* enhancer (*Chan et al., 2010*).

We investigated nucleotide diversity in 51 wild silkworms and 171 domesticated silkworms (*Tong et al., 2022*; *Lu et al., 2024*). The nucleotide diversity of introns and upstream sequences of *Bm-mamo* in wild silkworms was significantly greater than that in domestic silkworms. In addition, the approximately 1 kb genomic region upstream had high fixation indices ($F_{ST}$) (*Figure 12*). This indicates a significant degree of differentiation between the wild strains in this genomic region. Multiple sequence alignment analysis of this region was performed between 12 wild silkworms and 12 domestic silkworms (*Figure 12—figure supplement 1* and *Supplementary file 9–Table S9*). The results showed that the wild silkworm group had a greater degree of nucleotide variation, while the sequences in domestic silkworms were highly conserved. The domestic silkworm group had two different forms: one had a long interspersed nuclear element (LINE) inserted at approximately 4.5 kb, and the other

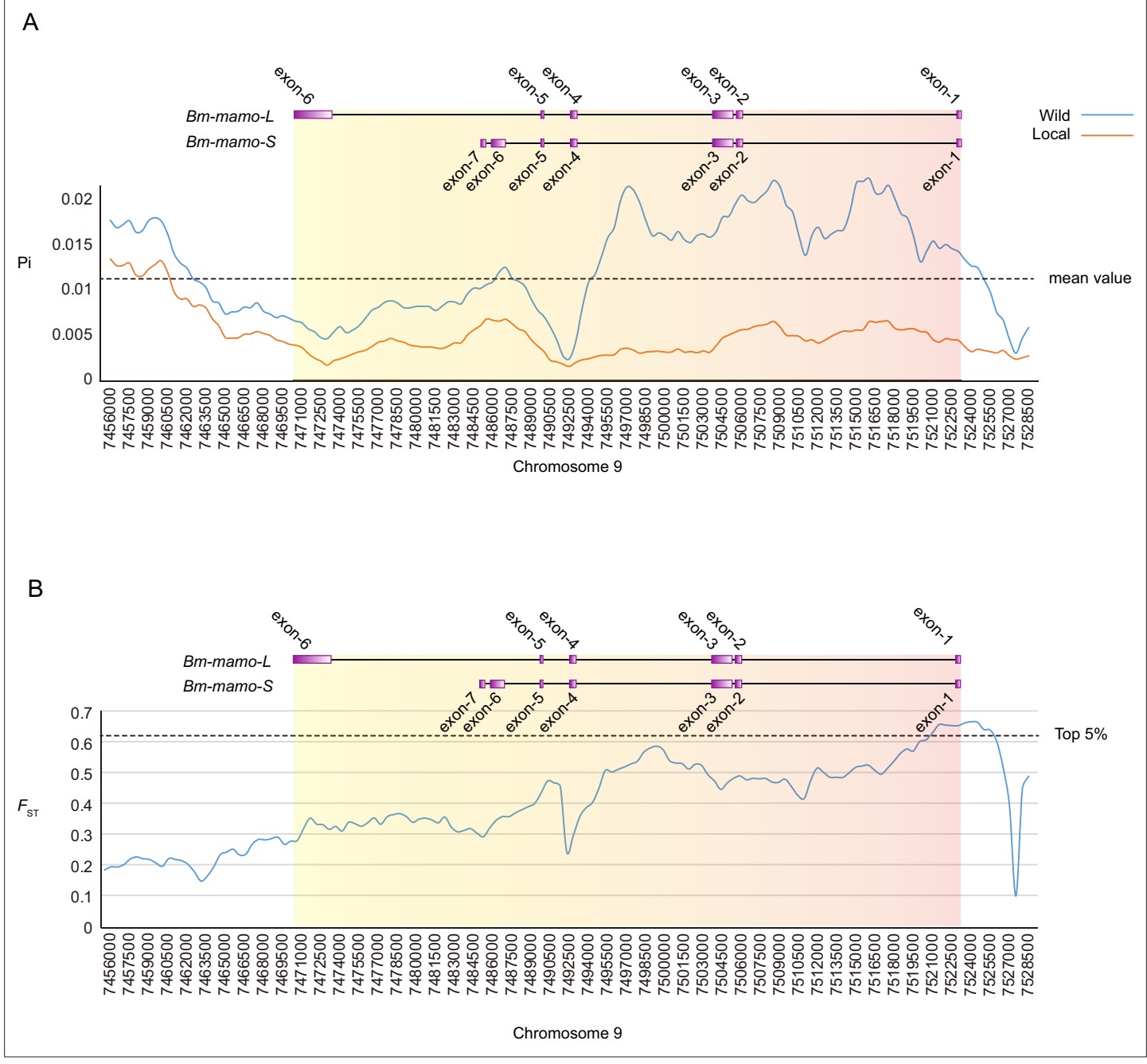

**Figure 12.** Investigation of nucleotide diversity and fixation index in the *Bm-mamo* genomic region using 51 wild and 171 domestic silkworm strains. (**A**) Nucleotide diversity analysis. The greater the value of pi (π) is, the greater the diversity of nucleotide sequences within the population. The dashed line represents the average nucleotide diversity level of the silkworm. The *X*-axis represents the position of chromosome 9. (**B**) Fixation index analysis. The greater the value of $F_{ST}$ is, the greater the degree of differentiation of alleles between populations. The dashed line represents the top 5% fixation level of silkworms. The *X*-axis represents the position of chromosome 9.

The online version of this article includes the following source data and figure supplement(s) for figure 12:

**Figure supplement 1.** The multiple sequence alignment of the upstream sequences of *Bm-mamo* among 12 wild silkworms and 12 domesticated silkworms.

**Figure supplement 1—source data 1.** Upstream sequences of *mamo* genes in 12 wild silkworm and 12 domestic silkworm strains.

did not have this transposon sequence. It is suggested that this genomic region is prone to variation in wild silkworm and is relatively fixed in domestic silkworms. The larvae of domesticated strains often have a lighter or even pale body color, but the wild-type strains have a darker color. The *Bm-mamo* gene may be involved in the domestication of silkworms.

TFs function by precisely regulating the temporal and spatial expression of target genes. Therefore, the expression of TFs requires strict regulation. There are long intergenic regions upstream of many important TFs, dozens of kilobase pairs (kb) to hundreds of kb, which may contain many CREs for better control of their expression pattern. It is often believed that changes in CREs are caused by random mutations. However, recent studies have shown that there is a mutation bias in the genome; compared with that in the intergenic region, the mutation frequency is reduced by half inside gene bodies and by two-thirds in essential genes. In addition, they compared the mutation rates of genes with different functions. The mutation rate in the coding region of essential genes (such as translation) is the lowest, and the mutation rates in the coding region of specialized functional genes (such as environmental response) are the highest. These patterns are mainly affected by the features of the epigenome (*Monroe et al., 2022*). Due to the plasticity of epigenomic features, mutation bias associated with epigenomes may even lead to environmental influences on mutations (*Belfield et al., 2021*). This finding suggested that different functional regions in the genome are subject to distinct controls and that some sequences can undergo procedural changes under environmental changes. Therefore, some variations in the CREs of TFs in response to environmental changes may be mutation bias.

## Materials and methods

### Silkworm strains

The *bd* and *bd*$^f$ mutant strains and the wild-type Dazao and N4 strains were obtained from the bank of genetic resources of Southwest University. Silkworms were reared on mulberry leaves or artificial diets at 25°C and 73% relative humidity in the dark for 12 hr and light for 12 hr.

### Positional cloning of *bd*

For mapping of the *bd* locus, F1 heterozygous individuals were obtained from a cross between a *bd*$^f$ strain and a Dazao strain. Then, an F1 female was crossed with a *bd*$^f$ male (BC1F), and an F1 male was backcrossed with a *bd*$^f$ female (BC1M). A total of 1162 BC1M individuals were used for recombination analysis. Genomic DNA was extracted from the parents (Dazao, *bd*$^f$, and F1) and each BC1 individual using the phenol chloroform extraction method. Available DNA molecular markers were identified through polymorphism screening and linkage analysis. The primers used for mapping are listed in *Supplementary file 10–Table S10*.

### Phylogenetic analysis

To determine whether *Bm-mamo* orthologs were present in other species, the BlastX program of the National Center for Biotechnology Information (NCBI) (http://www.ncbi.nlm.nih.gov/BLAST/) was used. The *Bm-mamo-L* and *Bm-mamo-S* sequences were subjected to BLAST searches against the nonredundant protein sequence (nr) database. Sequences with a maximum score and an *E*-value ≤10$^{-4}$ were downloaded. The sequences of multiple species were subjected to multiple sequence alignment of the predicted amino acid sequences by MUSCLE (*Edgar, 2004*). A phylogenetic tree was subsequently constructed using the neighbor-joining method with the MEGA7 program (Pearson model). The confidence levels for various phylogenetic lineages were estimated by bootstrap analysis (2000 replicates).

### Quantitative PCR

Total RNA was isolated from the whole body and integument of the silkworms using TRIzol reagent (Invitrogen, California, USA), purified by phenol chloroform extraction and then reverse transcribed with a PrimeScript RT Reagent Kit (TAKARA, Dalian, China) according to the manufacturer's protocol.

qPCR was performed using a CFX96 Real-Time PCR Detection System (Bio-Rad, Hercules, CA) with a qPCR system and an iTaq Universal SYBR Green Supermix System (Bio-Rad). The cycling parameters were as follows: 95°C for 3 min, followed by 40 cycles of 95°C for 10 s and annealing for 30 s. The primers used for the target genes are listed in *Supplementary file 10–Table S10*. The expression

levels of the genes in each sample were determined with biological replicates, and each sample was analyzed in triplicate. The gene expression levels were normalized against the expression levels of the ribosomal protein L3 (RpL3). The relative expression levels were analyzed using the classical $R = 2^{-\Delta\Delta Ct}$ method.

## siRNA for gene knockdown

siRNAs for *Bm-mamo* were designed with the siDirect program (http://sidirect2.rnai.jp). The target siRNAs and negative controls were synthesized by Tsingke Biotechnology Company Limited. siRNA (5 µl, 1 µl/µg) was injected from the abdominal spiracle into the hemolymph at the fourth-instar (Day 3) larval stage. Immediately after injection, phosphate-buffered saline (pH 7.3) droplets were placed nearby, and a 20-V pulse for 1 s and pause for 1 s were applied three times. The phenotype was observed for fifth-instar larvae. The left and right epidermis were separately dissected from the injected larvae, after which RNA was extracted. Then, cDNA was synthesized, and the expression level of the gene was detected via qPCR.

## Single-guide RNA synthesis and RNP complex assembly

CRISPRdirect (http://crispr.dbcls.jp/doc/) online software was used to screen appropriate single-guide RNA (sgRNA) target sequences. The gRNAs were synthetized by the Beijing Genomics Institute. The sgRNA templates were transcribed using T7 polymerase with RiboMAX Large-Scale RNA Production Systems (Promega, Beijing, China) according to the manufacturer's instructions. The RNA transcripts were purified using 3 M sodium acetate (pH 5.2) and anhydrous ethanol (1:30) precipitation, washed with 75% ethanol, and eluted in nuclease-free water. All injection mixtures contained 300 ng/µl Cas9 nuclease (Invitrogen, California, USA) and 300 ng/µl purified sgRNA. Before injection, mixtures of Cas9 nuclease and gRNA were incubated for 15 min at 37°C to reconstitute active RNPs (*Zou et al., 2022*).

## Microinjection of embryos

For embryo microinjection, microcapillary needles were manufactured using a PC-10 model micropipette puller (Narishige, Tokyo, Japan). Microinjection was performed using an Eppendorf TransferMan NK 2 and a FemtoJet 4i system (Eppendorf, Hamburg, Germany). The eggs used for microinjection were generated by mating female and male wild-type Dazao moths. Within 4 hr of culture, the eggs were allowed to adhere to a clean glass slide. CRISPR/Cas9-messenger RNP mixtures with volumes of approximately 1 nl were injected into the middle of the eggs, and the wound was sealed with glue. All the injected embryos were allowed to develop in a climate chamber at 25°C and 80% humidity (*Long et al., 2021*).

## Comparative genomics

The reference genome of Dazao was downloaded from Silkbase (https://silkbase.ab.a.u-tokyo.ac.jp/cgi-bin/index.cgi). The $bd^f$ genome was obtained from the silkworm pangenome project (*Tong et al., 2022*).

The short reads of the $bd^f$ strains were mapped to the silkworm reference genome by BWA55 v0.7.17 mem with default parameters. The SAMtools v1.11(https://github.com/samtools/samtools; *Danecek et al., 2024*) and Picard v2.23.5 (https://broadinstitute.github.io/picard/) programs were used to filter the unmapped and duplicated reads. A GVCF file of the samples was obtained using GATK57 v4.1.8.1 HaplotypeCaller with the parameter -ERC=GVCF. The VCF files of insertions/deletions (indels) and single-nucleotide polymorphisms were used for further analysis via eGPS software.

## Downstream target gene screening

Online software (http://zf.princeton.edu) was used to predict the DNA-binding site for Cys2His2 zinc finger proteins. The confident ZF domains with scores higher than 17.7 were chosen. RF regression on the B1H model was used to predict the DNA-binding sites. Then, the sequence logo and PWMs of the DNA-binding sites were obtained. The sequences 2 kb upstream and downstream of the predicted genes were extracted by a Perl script. The FIMO package of the MEME suite was used to search for binding sites in the silkworm genome.

## Analysis of EMSA

Primers with binding sites and flanking sequences were designed according to the FIMO results. A biotin label was added to one end of the upstream primer (probe), and the downstream primer was used as a reference. Primers with the same sequence as the labeled probes were used as competitive probes. The EMSA was conducted with an EMSA reagent kit (Beyotime, Shanghai, China) according to the manufacturer's protocol.

## Acknowledgements

This work was supported by the National Natural Science Foundation of China (Nos. 32002230, 32330102, and U20A2058), the Fundamental Research Funds for the Central Universities in China (No. SWU120024), National Key Research and Development Program (Project Nos. 2023YFD1600901 and 2023YFF1103801), the Natural Science Foundation of Chongqing, China (No. cstc2021jcyj-cxtt0005), and the High-level Talents Program of Southwest University (No. SWURC2021001).

# Additional information

## Funding

| Funder | Grant reference number | Author |
| --- | --- | --- |
| National Natural Science Foundation of China | 32002230 | Songyuan Wu |
| National Natural Science Foundation of China | 32330102 | Fangyin Dai |
| National Natural Science Foundation of China | U20A2058 | Fangyin Dai |
| Fundamental Research Funds for the Central Universities | SWU120024 | Songyuan Wu |
| National Key Research and Development Program of China | 2023YFD1600901 | Fangyin Dai |
| National Key Research and Development Program of China | 2023YFF1103801 | Fangyin Dai |
| Natural Science Foundation of Chongqing | cstc2021jcyj-cxtt0005 | Fangyin Dai |
| High-level Talents Program of Southwest University | SWURC2021001 | Fangyin Dai |

The funders had no role in study design, data collection, and interpretation, or the decision to submit the work for publication.

## Author contributions

Songyuan Wu, Data curation, Funding acquisition, Validation, Investigation, Visualization, Methodology, Writing - original draft, Writing - review and editing; Xiaoling Tong, Supervision, Writing - review and editing, Discussion; Chenxing Peng, Jiangwen Luo, Chenghao Zhang, Kunpeng Lu, Xin Ding, Xiaohui Duan, Yaru Lu, Hai Hu, Duan Tan, Investigation; Chunlin Li, Writing - review and editing, Discussion; Fangyin Dai, Resources, Supervision, Funding acquisition, Writing - review and editing

## Author ORCIDs

Songyuan Wu http://orcid.org/0000-0002-5732-8539
Xiaoling Tong http://orcid.org/0000-0002-2649-899X
Jiangwen Luo http://orcid.org/0009-0004-2084-0725
Fangyin Dai http://orcid.org/0000-0002-0215-2177

Joint Public Review: https://doi.org/10.7554/eLife.90795.4.sa1
Author response https://doi.org/10.7554/eLife.90795.4.sa2

## Additional files

### Supplementary files

• Supplementary file 1. $bd^f$ linkage analysis of the BC$_1$ population. 'A' indicates homozygosity, and 'H' indicates heterozygosity.

• Supplementary file 2. The indel and single-nucleotide polymorphism (SNP) in the region responsible for $bd^f$.

• Supplementary file 3. Binding site analysis of Bm-mamo-L. The position weight matrix of the mamo-L protein was used to search for 2000 base pair regions upstream and downstream of the predicted silkworm gene.

• Supplementary file 4. Binding site analysis of Bm-mamo-S. The position weight matrix of the mamo-s protein was used to search for 2000 base pair regions upstream and downstream of the predicted silkworm gene.

• Supplementary file 5. Differentially expressed genes between bd/bd and +/bd individuals.

• Supplementary file 6. The orthologs of 19 cuticular protein genes.

• Supplementary file 7. The genes encoding cuticular proteins with high expression in the black epidermal region of *Papilio xuthus* caterpillars. Futahashi R, et al. Comprehensive microarray-based analysis for stage-specific larval camouflage pattern-associated genes in the swallowtail butterfly, *Papilio xuthus*. BMC Biol. 2012;10:46. Published 2012 May 31. doi:10.1186/1741-7007-10-46

• Supplementary file 8. The temporal expression of the *mamo* gene in *Drosophila melanogaster*. Brown JB, Boley N, Eisman R, et al. Diversity and dynamics of the *Drosophila* transcriptome. Nature. 2014;512(7515):393–399. doi:10.1038/nature12962

• Supplementary file 9. Information on the silkworm strains. p indicates that the larva has a pale body color and has no markings. p3 indicates that the larva has a pale body color and has eye spots, lunar spots, and star spots. The hyphen indicates not available. The wild indicates the wild silkworm (*Bombyx mandarina*). 'Local' indicates domesticated strains.

• Supplementary file 10. Primers used for this research.

• MDAR checklist

### Data availability

Sequencing data have been deposited in GenBank under accession numbers PP426589 and PP426590. In additional, all data generated or analyzed during this study are included in the manuscript and supporting files; source data files have been provided for Figures 2, 6 , 9, and 11.

The following datasets were generated:

| Author(s) | Year | Dataset title | Dataset URL | Database and Identifier |
| --- | --- | --- | --- | --- |
| Wu S | 2024 | Bombyx mori strain Dazao mamo-L transcript 2 (mamo) mRNA, complete cds | https://www.ncbi.nlm.nih.gov/nuccore/PP426589.1/ | NCBI GenBank, PP426589 |
| Wu S | 2024 | Bombyx mori strain Dazao mamo-S transcript 1 (mamo) mRNA, complete cds | https://www.ncbi.nlm.nih.gov/nuccore/PP426590.1/ | NCBI GenBank, PP426590 |

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
