## [Editor Report · eLife assessment]

This **important** study identifies the gene mamo as a new regulator of pigmentation in the silkworm *Bombyx mori*, a function that was previously unsuspected based on extensive work on *Drosophila* where the *mamo* gene is involved in gamete production. The evidence supporting the role of *Bm-nano* in pigmentation is **convincing**, including high-resolution linkage mapping of two mutant strains, expression profiling, and reproduction of the mutant phenotypes with state-of-the-art RNAi and CRISPR knock-out assays. The work will be of interest to evolutionary biologists and geneticists studying color patterns and evolution of gene networks.

---

## [Referee Report · Joint Public Review]

This papers performs fine-mapping of the silkworm mutants bd and its fertile allelic version, bdf, narrowing down the causal intervals to a small interval of a handful of genes. In this region, the gene orthologous to mamo is impaired by a large indel, and its function is later confirmed using expression profiling, RNAi, and CRISPR KO. All these experiments are convincingly showing that mamo is necessary for the suppression of melanic pigmentation in the silkworm larval integument.

The authors also use in silico and in vitro assays to probe the potential effector genes that mamo may regulate.

The genotype-to-phenotype workflow, combining forward (mapping) and reverse genetics (RNAi and CRISPR loss-of-function assays) linking mamo to pigmentation are extremely convincing.

Comments on latest version:

This second revision took into account all the reviewers' comments. The authors added an interesting analysis of nucleotide diversity at the Bm-mamo locus, using available sequence data from 51 wild silkworms and 171 domesticated silkworms.

The last paragraph added to the discussion, starting with "It has often been believed that changes in CREs are caused by random mutations", is speculative. There is currently no evidence that the mutation rate is biased at the Bm-mamo locus.

---

## [Author Response]

The following is the authors’ response to the previous reviews.

eLife assessmentThis important study identifies the gene mamo as a new regulator of pigmentation in the silkworm Bombyx mori, a function that was previously unsuspected based on extensive work on *Drosophila* where the mamo gene is involved in gamete production. The evidence supporting the role of Bm-nano in pigmentation is convincing, including high-resolution linkage mapping of two mutant strains, expression profiling, and reproduction of the mutant phenotypes with state-of-the-art RNAi and CRISPR knock-out assays. While the discussion about genetic changes being guided or accelerated by the environment is extremely speculative and has little relevance for the findings presented, the work will be of interest to evolutionary biologists and geneticists studying color patterns and evolution of gene networks.

Response: Thank you very much for your careful work. In the revised version, we conducted a comparative genomic analysis of the upstream regions of the Bm-mamo gene in 51 wild silkworms and 171 domesticated local silkworms. The analysis of nucleotide diversity (pi) and the fixation index (FSTs) of the Bm-mamo genome sequences in the wild and domesticated silkworm populations were also performed. The results showed that the Bm-mamo genome sequence of local silkworms was relatively conserved, while the upstream sequence of wild silkworms exhibited high nucleotide diversity. This finding suggested a high degree of variability in the regulatory region of the Bm-mamo gene, in wild strains. Additionally, the sequence in this region may have been fixed by domestication selection. We have optimized the description in the discussion section.

**Public Reviews:**

**Reviewer #1 (Public Review):**
Summary:This papers performs fine-mapping of the silkworm mutants bd and its fertile allelic version, bdf, narrowing down the causal intervals to a small interval of a handful of genes. In this region, the gene orthologous to mamo is impaired by a large indel, and its function is later confirmed using expression profiling, RNAi, and CRISPR KO. All these experiments are convincingly showing that mamo is necessary for the suppression of melanic pigmentation in the silkworm larval integument.The authors also use in silico and in vitro assays to probe the potential effector genes that mamo may regulate.Strengths:The genotype-to-phenotype workflow, combining forward (mapping) and reverse genetics (RNAi and CRISPR loss-of-function assays) linking mamo to pigmentation are extremely convincing.This revision is a much improved manuscript and I command the authors for many of their edits.

Response: Thank you very much for your careful work. With the help of reviewers and editors, we have revised the manuscript to improve its readability.

I find the last part of the discussion, starting at "It is generally believed that changes in gene expression patterns are the result of the evolution of CREs", to be confusing.In this section, I believe the authors sequentially:emphasize the role of CRE in morphological evolution (I agree)emphasize that TF, and in particular their own CRE, are themselves important mutational targets of evolution (I agree, but the phrasing need to insist the authors are here talking about the CRE found at the TF locus, not the CRE bound by the TF).use the stickleback Pel enhancer as an example, which I think is a good case study, but the authors also then make an argument about DNA fragility sites, which is hard to connect with the present study.then continue on "DNA fragility" using the peppered moth and butterfly cortex locus. There is no evidence of DNA fragility at these loci, so the connection does not work. "The cortex gene locus is frequently mutated in Lepidoptera", the authors say. But a more accurate picture would be that the cortex locus is repeatedly involved in the generation of color pattern variants. Unlike for Pel fragile enhancer, we don't know if the causal mutations at this locus are repeatedly the same, and the haplotypes that have been described could be collateral rather than causal. Overall, it is important to clarify the idea that mutation bias is a possible factor explaining "genetic hotspots of evolution" (or genetic parallelism sensu 10.1038/nrg3483), but it is also possible that many genetic hotspots are repeated mutational targets because of their "optimal pleiotropy" (e.g. hub position in GRNs, such as mamo might be), or because of particularly modular CRE region that allow fine-tuning. Thus, I find the "fragility" argument misleading here. In fact the finding that "bd" and "bdf" alleles are different in nature is against the idea of a fragility bias (unless the authors can show increased mutation rates at this locus in a wild silkmoth species?). These alleles are also artificially-selected ie. they increased in frequency by breeding rather than natural selection in the wild, so while interesting for our understand of the genotype-phenotype map, they are not necessarily representative of the mutations that may underlie evolution in the wild.

Response: Thank you very much for your careful work. DNA fragility is an interesting topic, but some explanations for DNA fragility are confusing. One study measured the rate of DNA double-strand breaks (DSBs) in yeast artificial chromosomes (YACs), which are chromosomes containing marine Pel that broke ~25 to 50 times more frequently than did the control. These authors believe that the increase in the mutation rate is caused by DNA sequence characteristics, particularly TG-dinucleotide repeats. Moreover, they found that adding a replication origin on the opposite side of Pel did not cause the fungus to switch fragile, making the forward sequence stable and the reverse complement fragile. Thus, Pel fragility is also dependent on the direction of DNA replication. In summary, they suggested that the special DNA sequence is the cause of DNA fragility. In addition, the sequence features associated with DNA fragility in the Pel region are also found in thousands of other positions in the stickleback and human genomes (Xie KT et al, 2019, science).

In yeast artificial chromosomes (YACs), the characteristics of DNA sequences, such as TG-dinucleotide repeat sequences, may be important reasons for DNA fragility, and these breaks occur during DNA replication. However, the inserted sequence of YAC often undergoes deletion or recombination during cultivation and passage. In addition, yeast is a single-celled organism. Therefore, the results in yeast cannot represent the situation in multicellular organisms. If multicellular organisms are like this, there are several issues as follows:

(1) The DNA replication process occurs separately in different multicellular organisms. Because DNA breakage and repair are independent, they can lead to the presence of different alleles in different cells. This can potentially lead to the occurrence of extensive chimeric organisms. However, we have not found such a situation in the genome sequencing of many multicellular organisms.

(2) If the DNA sequence, TG-dinucleotide repeats, is the determining factor, the mutations near the sequence lose their strong correlation with environmental changes. The researchers conducted yeast artificial chromosome experiments in the same environment and found that the frequency of DNA breaks containing TG dinucleotide repeat sequences was 25 to 50 times greater than that of the control group. This means that, whether in the marine population or the lake population, this part of the sticklebacks’ genome has undergone frequent mutations. However, according to related research, populations of lake sticklebacks, rather than marine populations, often exhibit a decrease in the pelvic phenotype.

(3) Researchers have found thousands of loci in the genome of sticklebacks and humans that contain such sequences (TG-dinucleotide repeats). This means that thousands of sites undergo frequent mutations during DNA replication. Unless these sites do not possess functionality, they will have some impact on the organism, even causing damage. Even if they are not functional sequences, these sequences will gradually be discarded or replaced during frequent mutations rather than being present in large quantities in the genome.

Therefore, the study of DNA fragility in yeast cannot explain the situation in multicellular organisms.

As you noted, we want to express that the frequent variation in the cortex gene should be regulated by targeted regulation involving the GRN in Lepidoptera. In addition, studies on specific epigenetic modifications discovered through the referenced fragile DNA sites suggest that DNA fragility is not determined by the DNA sequence (Ji F, 2020, Cell Res) but rather by other factors, such as epigenetic factors. The sequence features discovered at fragile DNA sites are traces of frequent mutations, not causes.

In this revision, we analyzed the nucleotide diversity of the mamo genome in 51 wild and 171 domestic silkworms. We found high nucleic acid diversity from the third exon to the upstream region of this gene in wild silkworms. We randomly selected 12 wild silkworms and 12 domestic silkworms and compared their upstream sequences to approximately 1 kb. In wild silkworms, there is significant diversity in their upstream sequences. In domestic silkworms, the sequences are highly conserved, but in some silkworms, a long interspersed nuclear element (LINE) is inserted. This finding suggested that there is frequent variation in the sequence of this region in wild silkworms, while fixation occurs in domesticated silkworms. These genomic data are sourced from the pangenome of silkworms (Tong X, 2022, Nat Commun.). In the pangenomic research, 1078 strains (205 local strains, 194 improved strains, 632 mutant strains, and 47 wild silkworms), which included 545 third-generation sequencing genomes, were obtained. An online website was built to utilize these data (http://silkmeta.org.cn/). We warmly welcome you to use these data.

In summary, for clearer expression, we have rewritten this section.

Xie KT, Wang G, Thompson AC, Wucherpfennig JI, Reimchen TE, MacColl ADC, Schluter D, Bell MA, Vasquez KM, Kingsley DM. DNA fragility in the parallel evolution of pelvic reduction in stickleback fish. Science. 2019 Jan 4;363(6422):81-84. doi: 10.1126/science.aan1425.

Ji F, Liao H, Pan S, Ouyang L, Jia F, Fu Z, Zhang F, Geng X, Wang X, Li T, Liu S, Syeda MZ, Chen H, Li W, Chen Z, Shen H, Ying S. Genome-wide high-resolution mapping of mitotic DNA synthesis sites and common fragile sites by direct sequencing. Cell Res. 2020 Nov;30(11):1009-1023. doi: 10.1038/s41422-020-0357-y.

Tong X, Han MJ, Lu K, Tai S, Liang S, Liu Y, Hu H, Shen J, Long A, Zhan C, Ding X, Liu S, Gao Q, Zhang B, Zhou L, Tan D, Yuan Y, Guo N, Li YH, Wu Z, Liu L, Li C, Lu Y, Gai T, Zhang Y, Yang R, Qian H, Liu Y, Luo J, Zheng L, Lou J, Peng Y, Zuo W, Song J, He S, Wu S, Zou Y, Zhou L, Cheng L, Tang Y, Cheng G, Yuan L, He W, Xu J, Fu T, Xiao Y, Lei T, Xu A, Yin Y, Wang J, Monteiro A, Westhof E, Lu C, Tian Z, Wang W, Xiang Z, Dai F. High-resolution silkworm pan-genome provides genetic insights into artificial selection and ecological adaptation. Nat Commun. 2022 Sep 24;13(1):5619. doi: 10.1038/s41467-022-33366-x.

Lu K, Pan Y, Shen J, Yang L, Zhan C, Liang S, Tai S, Wan L, Li T, Cheng T, Ma B, Pan G, He N, Lu C, Westhof E, Xiang Z, Han MJ, Tong X, Dai F. SilkMeta: a comprehensive platform for sharing and exploiting pan-genomic and multi-omic silkworm data. Nucleic Acids Res. 2024 Jan 5;52(D1):D1024-D1032. doi: 10.1093/nar/gkad956.

Curiously, the last paragraph ("Some research suggests that common fragile sites...") elaborate on the idea that some sites of the genome are prone to mutation. The connection with mamo and the current article are extremely thin. There is here an attempt to connect meiotic and mitotic breaks to Bm-mamo, but this is confusing: it seems to propose Bm-mamo as a recruiter of epigenetic modulators that may drive higher mutation rates elsewhere. Not only I am not convinced by this argument without actual data, but this would not explain how the mutations at the Bm-mamo itself evolved.

Response: Thank you very much for your careful work. This section mainly illustrates that DNA fragility is not determined by sequence but is regulated by other factors in animals. In fruit flies, they found that mamo is an important candidate gene for recombination hotspot setting in meiosis. First, we evaluated PRDM9, which plays an important role in setting recombination hotspots during meiosis. Our purpose in mentioning this information is to illustrate that chromosome recombination is a process of programmed double strand breaks and to answer another reviewer's question about programmed events in the genome. In summary, we suggest that some variations in DNA sequences are procedural results.We have optimized the description of this section in this version.

On a more positive note, I find it fascinating that the authors identified a TF that clearly articulates or orchestrate larval pattern development, and that when it is deleted, can generate healthy individuals. In other words, while it is a TF with many targets, it is not too pleiotropic. This idea, that the genetically causal modulators of developmental evolution are regulatory genes, has been described elsewhere (e.g. Fig 4c in 10.1038/s41576-020-0234-z, and associated refs). To me, the beautiful findings about Bm-mamo make sense in the general, existing framework that developmental processes and regulatory networks "shape" the evolutionary potential and trajectories of organisms. There is a degree of "programmability" in the genomes, because some loci are particularly prone to modulate a given type of trait. Here, Bm-mamo, as a potentially regulator of both CPs and melanin pathway genes, appear to be a potent modulator of epithelial traits. Claiming that there are inherent mutational biases behind this is unwarranted.

Response: Thank you very much for your careful work. I completely agree with your statement that the genome exhibits a certain degree of programmability. On the one hand, some transcription factors can precisely control the spatiotemporal expression levels of some structural genes (such as pigment synthesis genes). On the other hand, these transcription factors are also subject to strict expression regulation. Because the color pattern is complex, changes in single or minority structural genes result in incomplete or imprecise changes in coloring patterns. Nevertheless, several regulatory factors can regulate multiple downstream target genes. Changes in their expression patterns can lead to holistic and significant changes in color patterns. There are long intergenic regions upstream of many important transcription factors, dozens of kilobase pairs (Kb) to hundreds of Kb, which may contain many different regulatory elements for better control of their expression patterns. Therefore, gene regulatory networks can directly regulate transcription factors to modulate a given type of trait. Transcription factors and their downstream target genes can form a functional module, which is similar to a functional module in software or operating systems. This regulation of transcription factors is simpler in terms of steps, which are similar to a single click switch button. The gene regulatory network regulates these modules in response to environmental changes and is widely recognized.

Some people do not agree that genetic variations can also be regulated. They claim that this is completely random. The infinite monkey theorem (Félix-Édouard-Justin-Émile Borel, 1909) states that if an infinite number of monkeys were given typewriters and an infinite amount of time, they would eventually produce the complete works of Shakespeare. Although this theory advocates randomness on the surface, its conclusions are full of inevitability (tail event). In nature, some things we observe do not have obvious regularity because they involve relatively complex factors, and the underlying logic is obscure and difficult to understand. We often name them random. However, as we gradually understand the logic behind this complex event, we can also recognize the procedural nature of this randomness.

Previously, chromosomal recombination during meiosis was believed to be a random event. However, currently, it is believed that the process is procedural. The occurrence of meiotic recombination mentioned earlier indicates that the genome has the ability to self-set the position of double-strand breaks to form new allelic forms. Because meiotic recombination is programmed, transcription factors that recognize DNA sites, enzymes that cleave double strands, and DNA repair systems exist, programming can also introduce genetic variation. A study in plants has provided insights into this programmed mutation (Monroe JG, 2023, nature).Frequent changes in the expression patterns of some transcription factors occur between and/or within species. In this article, we only discuss the possible reasons for variations in the expression patterns of some transcription factors in a general manner and simple reasoning. We have added an analysis of the response of wild silkworms and improved the relevance of the discussion.

Monroe JG, Srikant T, Carbonell-Bejerano P, Becker C, Lensink M, Exposito-Alonso M, Klein M, Hildebrandt J, Neumann M, Kliebenstein D, Weng ML, Imbert E, Ågren J, Rutter MT, Fenster CB, Weigel D. Mutation bias reflects natural selection in *Arabidopsis thaliana*. Nature. 2022 Feb;602(7895):101-105. doi: 10.1038/s41586-021-04269-6. Epub 2022 Jan 12. Erratum in: Nature. 2023 Aug;620(7973):

**Recommendations for the authors:**

**Reviewer #1 (Recommendations For The Authors):**
Please structure your Discussion with section headers.

Response: Thank you very much for your careful work. We have added relevant section headers.

As explained in my public review, I found the two last sections of the Discussion to be dispersed and confusing. I also must say that I carefully read the Response to Reviewers on this, which helped me to better understand the authors' intentions here. Please consider the revision of this Discussion as this feels extremely speculative difficult to connect with Bm-mamo.

Response: Thank you very much for your careful work. We have rewritten this part of the content.

typo: were found near the TTS of yellow  TSS

Response: Thank you very much for your careful work. We have made these modifications.

l. 234 :"expression level of the 18 CP genes in the integument". Consider adding a mention of Figure 7 here, as only Fig. S10 is cited here.

Response: Thank you very much for your careful work. We have made these modifications.

Editorial comment on the second half of the Abstract:Wu et al : "We found that Bm-mamo can comprehensively regulate the expression of related pigment synthesis and cuticular protein genes to form color patterns. This indicates that insects have a genetic basis for coordinate regulation of the structure and shape of the cuticle, as well as color patterns. This genetic basis provides the possibility for constructing the complex appearances of some insects. This study provides new insight into the regulation of color patterns."I respectfully suggest a more accurate rephrasing, where the methods are mentioned, and where the logical argument is more straightforward. For example"Using RNAi and CRISPR we show that Bm-mamo is a repressor or dark melanin patterns in the larval epithelium. Using in-vitro binding assays and gene expression profiling in wild-type and mutant larvae, we also show that Bm-mamo likely regulate the expression of related pigment synthesis and cuticular protein genes in a coordinated manner to mediate its role in color pattern formation. This mechanism is consistent with a dual role of this transcription factor in regulating both the structure and shape of the cuticle and pigments that are embedded within it. This study provides new insight into the regulation of color patterns as well as in the construction more complex epithelial features in some insects."I hope this let the ideas of the original version transpire as the authors intended.

Response: Thank you very much for your careful work. We have made these modifications.